# Defining the cell and molecular origins of the primate ovarian reserve

Sissy E. Wamaitha[1,2,3], Ernesto J. Rojas [4,5], Francesco Monticolo[6], Fei-man Hsu[1,2,3], Enrique Sosa[1,2,3], Amanda M. Mackie [1], Kiana Oyama[7], Maggie Custer[7], Melinda Murphy[7], Diana J. Laird [4,5], Jian Shu [6,8], Jon D. Hennebold[7,9] & Amander T. Clark [1,2,3] ✉

The primate ovarian reserve is established during late fetal development and consists of quiescent primordial follicles in the ovarian cortex each composed of granulosa cells surrounding an oocyte in dictate. As late stages of fetal development are not routinely accessible using human tissues, the current study exploits the evolutionary proximity of the rhesus macaque to investigate follicle formation in primates. Like in humans, the rhesus prenatal ovary develops multiple types of pre-granulosa cells in time and space, with primordial follicles deriving from later emerging pre-granulosa subtypes. In addition, our work shows that activated medullary follicles recruit fetal theca cells to establish a two-cell system for sex-steroid hormone production prior to birth, providing a cell-based explanation for mini puberty.

Ovarian follicles are the functional unit of the ovary, composed of a single dictate arrested oocyte surrounded by granulosa cells[1]. The most immature follicles are referred to as primordial follicles. These make up the ovarian reserve, and upon activation are essential for sex-hormone production and fertility. In primates, primordial follicles are established in late gestation; in humans, these are first observed around week 19–20 post conception (W19–20)[2,3] and around W17–20 in the rhesus macaque (*Macaca mulatta*)[2]. After birth, the infant pituitary gland begins to produce follicle-stimulating hormone (FSH) and luteinizing hormone (LH), which stimulate estrogen and anti-Mullerian hormone (AMH) production by the infant ovary[4–6]. This creates a state referred to as mini puberty, a critical period for brain and body development lasting from birth to around 6 months[7,8]. For those born with fewer primordial follicles (such as in Turner Syndrome) or with differences in sex development, mini puberty may not occur normally. This creates a window of opportunity for diagnosis and in some cases treatment for reproductive disease. Furthermore, as the ovarian reserve does not self-renew, the generation of fewer follicles prior to birth is also associated with a faster decline in sex-hormone production, primary ovarian insufficiency, and ultimately infertility[9]. Therefore, the establishment of ovarian follicles in late gestation is an understudied area of science with tremendous impact, given the importance of the ovary to overall health.

Unlike humans and nonhuman primates, primordial follicle formation in mice occurs after birth within the first few days of life[10]. Mice subsequently undergo a mini puberty from day 10–17, which involves follicle activation in response to FSH; these are called first-wave follicles. Blocking mini puberty is associated with reproductive anomalies, including delayed puberty onset in females and reproductive dysfunction[11]. First wave follicles in mice originate from an oocyte surrounded by bipotential pre-granulosa (BPG) cells[12]. In contrast, quiescent primordial follicles of the mouse ovarian reserve originate from

[1]Department of Molecular, Cell and Developmental Biology, University of California Los Angeles, Los Angeles, CA, USA. [2]Eli and Edythe Broad Center of Regenerative Medicine and Stem Cell Research, University of California, Los Angeles, Los Angeles, CA, USA. [3]Molecular Biology Institute, University of California, Los Angeles, Los Angeles, CA, USA. [4]Department of Obstetrics, Gynecology and Reproductive Science, Center for Reproductive Sciences, University of California, San Francisco, San Francisco, CA, USA. [5]Eli and Edythe Broad Center for Regeneration Medicine and Stem Cell Research, University of California, San Francisco, San Francisco, CA, USA. [6]Cutaneous Biology Research Center, Massachusetts General Hospital, Harvard Medical School, Boston, MA, USA. [7]Division of Reproductive & Developmental Sciences, Oregon National Primate Research Center, Beaverton, OR, USA. [8]Klarman Cell Observatory, Broad Institute of MIT and Harvard, Cambridge, MA, USA. [9]Department of Obstetrics and Gynecology, Oregon Health & Science University, Portland, OR, USA. ✉ e-mail: clarka@ucla.edu

oocytes that become enclosed by a second wave of cells called epithelial pre-granulosa (EPG) cells. Recent analysis in humans identified both BPG and EPG cognates, designated preGC-I (PG1) and preGC-IIa/b (PG2)[13,14]. Given the challenges in studying human prenatal tissue after 20 weeks, the fates of PG1 and PG2 with respect to follicle formation are unknown; discerning this will be critical to understanding fetal origins of ovarian disease and identifying ovarian cell types involved in mini puberty. As previous work has shown the sequence of gonadal developmental events in the rhesus macaque is comparable to the human[2], we investigated ovarian development at a cellular and molecular level, following emergence from the gonadal ridge through to follicle establishment. We used spatial and single cell transcriptomics coupled with fluorescent microscopy to determine the nature, position, and progression of germline and somatic cells during ovarian development and patterning, and the fate of the two major types of pre-granulosa (PG) cells during primate fetal follicle development.

## Results

### Mapping primate fetal ovarian development

Studies of early gonadal development describe similar morphological events in humans, rhesus, and cynomolgus macaques[2,3,13–20].

Embryonic ovary development (up to Carnegie Stage (CS) 23, W8 post conception in primates) begins with formation of the gonadal ridge (humans, W5; rhesus, cynomolgus W4), which is colonized by migrating germ cells before sex determination (humans, W6; rhesus, cynomolgus W5). The fetal ovary then expands rapidly; germline cells proliferate and by W9, human germline cells initiate meiotic entry. As the germline cells proliferate, they assemble into ovarian cords (nests) alongside somatic granulosa cells, which are separated from the stromal cells in the interstitium by a basement membrane (humans, W12–13; rhesus W12). These nests then break down, and by the time of birth individual oocytes in the cortex become surrounded by a layer of squamous granulosa to establish primordial follicles (human W19–20; rhesus W17–20; cynomolgus by W21) (Fig. 1A). Here we sought to collect comprehensive molecular data during ovarian development in the rhesus to investigate the emergence and fate of granulosa subtypes and formation of ovarian follicles.

Using the Oregon National Primate Research Center (ONPRC) time-mated breeding program (Supplementary Figs. 1 and 2 and Supplementary Table 1), we collected rhesus fetal tissue at key timepoints; W5, coincident with sex determination (D34, CS16, $n = 2$ biological replicates); W6, early gonadal expansion (D41, CS20,

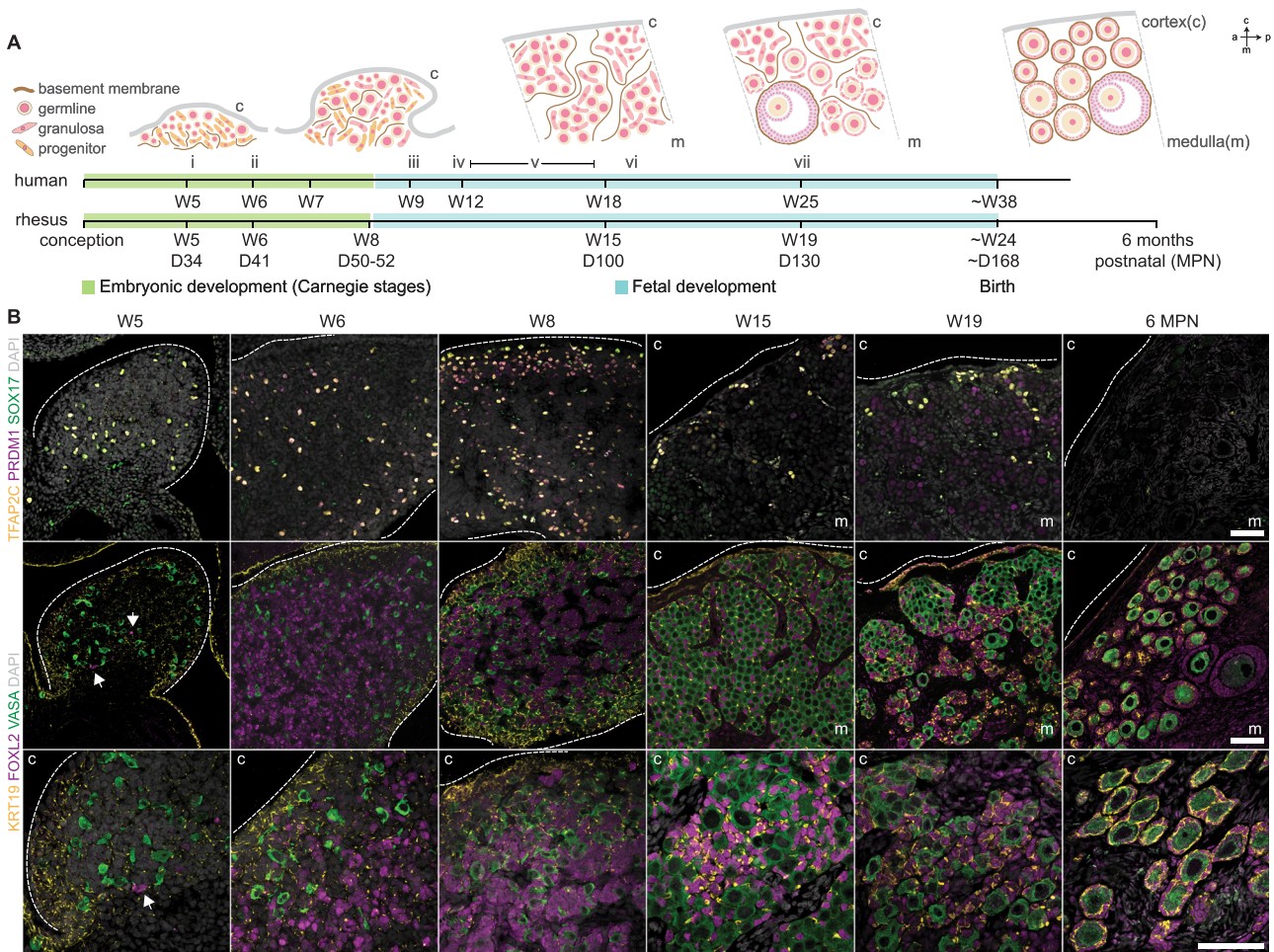

**Fig. 1 | Dynamic changes to the ovary precede follicle formation. A** Illustration of key events and morphological transitions during human and rhesus macaque ovarian development. Cortex (c), medulla (m), anterior (a), and posterior (p) labels show section image orientation—all subsequent images are positioned according to these axes. Roman numerals refer to key events: i—definitive ovary formation from gonadal ridge; ii—sex determination; iii—germline cells enter meiosis (leptotene oocytes observed); iv—ovigerous cord formation; v—diplotene oocytes observed; vi—primordial follicles emerge; vii—early activated follicles observed.

**B** Immunofluorescence analysis in W5 (D34) ($n = 2$), W6 (D41) ($n = 2$), W8 (D50-52) ($n = 3$), W15 (D100) ($n = 3$), W19 (D130) ($n = 4$) and 6 months postnatal rhesus ovaries ($n = 1$) for primordial germ cell (SOX17, green; PRDM1, magenta; TFAP2C, yellow); late-stage germ cell (VASA, green) and granulosa (KRT19, yellow; FOXL2, magenta) markers (bottom panels at higher magnification in the developing cortex). Arrows show rare FOXL2+ cells at W5. Nuclei counterstained with DAPI (grey). All scale bars 50 μM.

$n = 2$); W8, end of embryonic period and beginning of cord formation (D50–52, CS23, $n = 3$); W15, meiosis and nest expansion (D100, $n = 3$); W19, nest breakdown and primordial follicle formation (D130, $n = 3$); and 6 months, 6MPN ($n = 1$). All ages were calculated post conception (see "Methods"); birth in the rhesus has been reported at an average of 24 weeks (168 days), albeit with a wide range[21–23]. To track these morphological events, we first performed immunofluorescence analysis for known granulosa and germline markers identified in human, mouse, cynomolgus macaque, and bovine ovarian studies[12,14,18,24–26].

In the mouse, LGR5–/FOXL2 + BPG cells are derived from interior ovarian progenitor cells[12,27–29], while LGR5 + EPG cells originate from the KRT19+ ovarian surface epithelium and only acquire FOXL2 expression after birth[12]. This correlation infers that EPG-containing follicles are positively selected for survival. FOXL2 expression in the human ovary has been detected from as early as W6, coincident with sex determination and formation of PG1 cells[13,30]. FOXL2 + PG cells persist past W19[31], though it is unclear whether they originate from PG1 or PG2. In the rhesus, we found rare FOXL2+ cells in one ovarian replicate at W5 (W5_4, arrows Fig. 1B) but not the other (W5_3), consistent with its role as an early sex determination marker and likely expressed by early somatic progenitors and bipotential early supporting gonadal cells (ESGCs) akin to those first identified in humans at W6[13]. KRT19 expression was detected at W5, preceding FOXL2 expression (Fig. 1B). At W6 and W8, FOXL2+ cells were present in both ovarian center and cortex, and within ovarian cords at W15 and W19 (Fig. 1B). KRT19 was enriched at the surface epithelium at W6 and W8, consistent with its role as an epithelial marker, but was also expressed on some cortical and medullary FOXL2+ cells. By W15, all FOXL2+ cells within the cords were KRT19+, and this was maintained through follicle establishment at W19, with brighter KRT19 expression seemingly related to connection with neighboring pre-granulosa cells.

Primordial germ cell (PGC) markers SOX17, TFAP2C, and PRDM1[32] were initially detected in cells dispersed throughout the ovary at W5-8, but gradually restricted to the outer cortex by W15 and undetectable at 6MPN (Fig. 1B). The later germline marker VASA was detected at all stages, and VASA+ oocytes in primordial follicles were observed at W19 and 6MPN, surrounded by FOXL2 + KRT19+ granulosa cells (Fig. 1B). PRDM1 was also expressed in some W19 germline cells but was absent by 6MPN, which may indicate a transient role in the ovarian germline; in mice, Prdm1 CKO from E11.5 (post gonadal colonization) results in dramatic germline apoptosis after E16.5, likely linked to an aberrant pachytene phenotype[33]. Altogether, we found that FOXL2 + PG cells are present throughout the ovary by W6 onwards, with KRT19+/FOXL2 + PG cells being the predominant PG cells just prior to follicle formation.

## PG cells emerge from mesenchymal-like progenitors at W5–6

In humans, FOXL2 + PG cells are thought to differentiate from early somatic progenitors, which share expression with undifferentiated gonadal interstitial cells[13], expressing stromal-associated markers such as orphan nuclear receptor COUP-TFII/NR2F2[14]. In the rhesus, NR2F2 was expressed by the majority of cells in the budding gonad at W5–6, though appearing dimmer than those in the underlying mesenchyme (Supplementary Fig. 3A). NR2F2 showed substantial overlap with FOXL2 within the cords, with NR2F2+ interstitial stromal cells interspersed between them. By W8, NR2F2 was restricted to cells in the ovarian center, and ultimately to stromal cells outside the cords by W15. The extracellular matrix protein decorin was expressed only by NR2F2+ stromal cells, suggesting this could distinguish cells with a "true" ovarian stromal identity from NR2F2 + FOXL2+ gonadal PG progenitor cells (Supplementary Fig. 3A). NR2F2+ decorin+ stromal cells began infiltrating the fetal ovarian cortex coincident with nest breakdown at W19 and spanned the whole ovary by 6 MPN. The extracellular matrix protein laminin exhibited discontinuous

deposition in the W5 ovary, in contrast to the contiguous membrane visible on nearby tissues (Supplementary Fig. 3A, stars). Similar to the human, at W6–8 laminin marked the boundary between the cords and surrounding stromal cells, and was expressed within the expanding germline nests from W8 and by oocytes in primordial follicles at 6MPN.

Although much is known about the cell and molecular development of the human and cynomolgus macaque ovaries during the embryonic stages, very little is known about the rhesus. To identify a transcriptional signature of the rhesus gonadal primordia relative to neighboring organs derived from the intermediate mesoderm such as adjacent adrenal and kidney (mesonephric) tissues, we first performed spatial transcriptomics at W5 and W6 using the 10× Genomics Visium CytAssist platform with the human whole transcriptome probe set. Based on homology with the rhesus genome, we anticipated ~82% on-target coverage (Supplementary Fig. 3B; see "Methods"). Though the 55 µM Visium spots were larger than individual cells at this stage, this has advantages over bulk RNA-seq with respect to multi-organ comparison in a single embryo. At W5 and W6, embryonic ovaries were enriched for genes associated with indifferent gonads in mice, cynomolgus macaques and humans[18,34] (LHX9, NR5A1 (SF-1), WNT6, EMX2, NROB1 (DAX1)), as well as germline, granulosa and stromal cell identity (DDX4 (VASA), AMHR2, NR2F2) (Fig. 2A, B and Supplementary Fig. 3C, D). NR5A1 and NROB1 were also found in the adrenal gland along with CYP17A1. GATA2 was enriched in the mesonephros, and EMX2 was expressed in both the mesonephros and the metanephros, consistent with previous results[13,35–37]. RUNX1, which has been implicated in ovarian development, was detected in the ovary at these early developmental stages[38], while PAX8 mRNA was abundant in the mesonephros, and lightly detected in adjacent ovarian regions, possibly reflecting emerging rete ovarii progenitors[13,39,40].

We performed differential gene expression analysis to compare the ovary to the tissues visible in the section, which include the adrenal gland and mesonephros, as well as metanephros, hindgut, liver, aorta, spinal cord and abdominal muscles (apart from W6_1, as adrenal tissue was not present in the section plane) (Supplementary Fig. 1 and Figs. 2A, B and 3C, D). In addition to known enriched genes (FOXL2, WT1, GATA4, WNT6, AMHR2, LHX9), we identified GATM, a glycine aminotransferase involved in creatine biosynthesis that is also present in human fetal gonads[40]; GREB1, an estrogen receptor-regulated gene associated with hyperproliferation in ovarian and breast cancers[41,42]; and DUOX2, a reactive oxygen species-producing oxidase[43]. GREB1 knockout mice are sub-fertile, while overexpression in ovarian cancer cell lines promotes a mesenchymal morphology[42,44]. We also identified MBNL3 and IFI6, which are associated with proliferation, survival, and apoptotic resistance, and MFGE8, an integrin-binding protein present in mouse stromal and goat granulosa cells[45,46].

The steroidogenic enzyme GSTA3 was detected in both the ovary and adrenal primordium at W6; it functions downstream of NR5A1 and is involved in the HSD3B steroid hormone biosynthesis pathway[47]. Additional steroidogenic pathway components CYP17A1, CYP11A1, and STAR were enriched in the adrenal region, while GATA2, INHBA, and LUM were expressed in the mesonephros along with SULT1E1, an estrogen sulfotransferase also detected in human fetal mesonephric stromal cells[40,48] (Fig. 2A, B, Supplementary Fig. 3C, D, and Supplementary Data 1). The protein-tyrosine phosphatase PTPRO (GLEPP1), stomatin NPHS2 (podocin), and REN (renin), which are required for kidney function, were also enriched in the mesonephros[49,50] (Supplementary Data 1).

Comparing W5 to W6 ovaries only yielded a handful of differentially expressed genes (DEGs) (Supplementary Data 1). Comparing W5_3 (FOXL2–) and W5_4 (FOXL2+) ovaries did not identify any differentially expressed genes that met our thresholds at this stage. We collected equivalent testis samples at W5 ($n = 2$) and W6 ($n = 1$) to compare to W5 and W6 ovaries; as with the W5 ovaries, one testis had initiated SOX9 expression (W5_1), while the other had not (W5_2). We

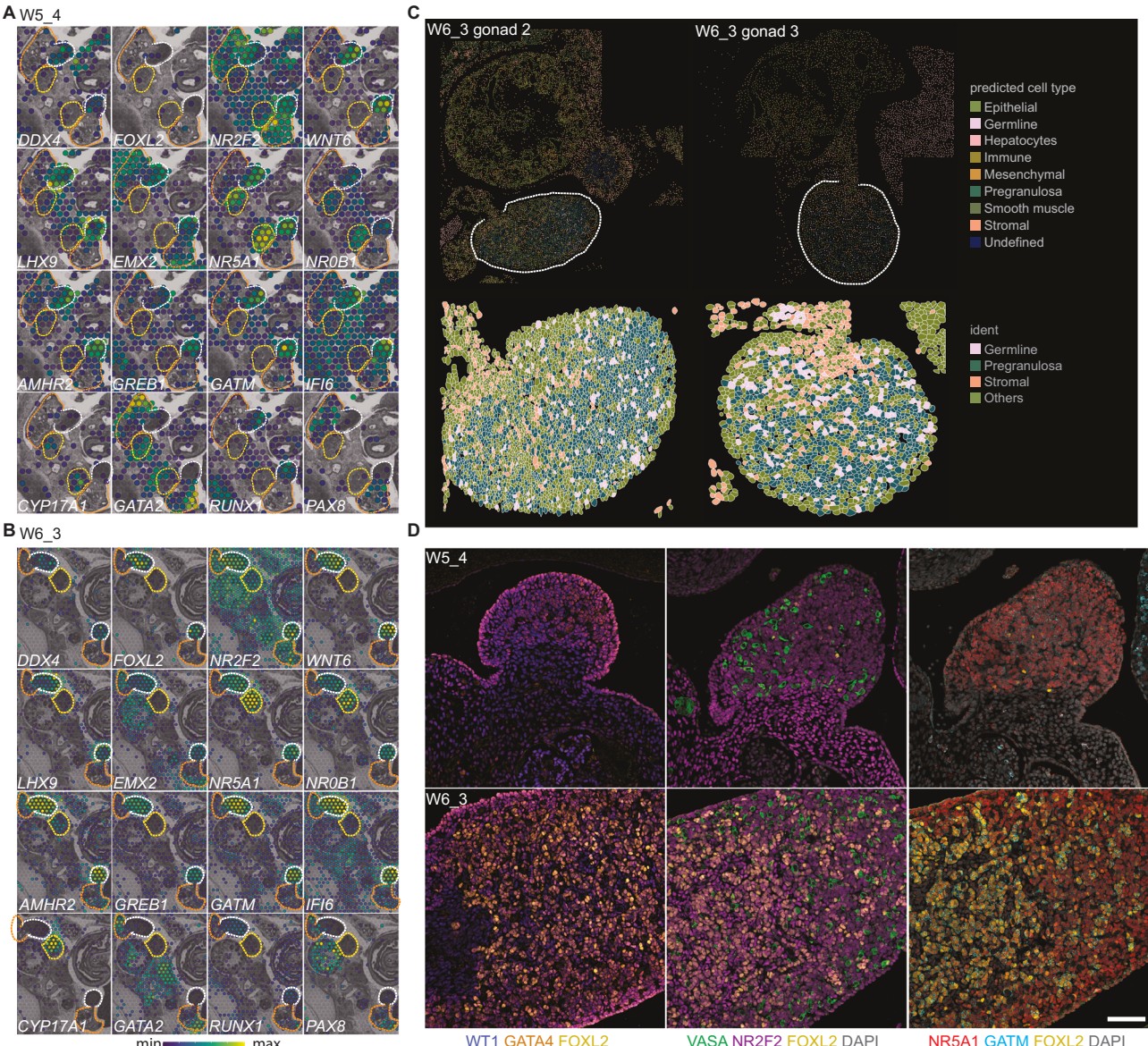

**Fig. 2 | Pre-granulosa (PG) cells emerge from mesenchymal-like progenitors at W5–6. A, B** Spatial distribution of genes of interest in Visium CytAssist analysis of W5 and W6 ovaries mapped onto a black and white H&E image. Ovary (white), mesonephros (orange), and adrenal (yellow) regions are outlined; expression level scaled from blue (min) to yellow (max). **C** CosMx spatial molecular imager (SMI) analysis of W6_3 ovaries; left panels, distribution of annotated cell clusters in the selected fields of view (FOVs); right panels, zoom in on ovary region with cell segmentation overlay showing germline, pre-granulosa and stromal cells (additional clusters marked "Others"). White dashed line indicates ovary region. **D** Immunofluorescence analysis for known genes (WT1, blue; GATA4, orange; FOXL2, yellow; VASA, green; NR2F2, magenta) and genes identified in the DEG analysis (NR5A1, red; GATM, cyan) in W5 (*n* = 2) and W6 (*n* = 2) ovaries. Scale bars 50 μM.

did not identify any significant DEGs between W5 indifferent gonads (W5_2 testis vs. W5_3 ovary), or between the W6 testis and W5 ovaries (Supplementary Data 1). Comparing the W6 testis to W6 ovaries identified only a handful of DEGs (Supplementary Data 1). Altogether, we identified a broadly consistent ovarian signature at W5–6 that is distinct from the emerging adrenal gland and mesonephros but retains global similarity to the emerging testis.

To localize gene expression to single cells, we performed Nano-String® CosMx™ Spatial Molecular Imaging (SMI) on W6 ovaries, using the 1K human gene panel. Cluster analysis identified 15 clusters in the W6_3 tissue section; cell types were assigned using known marker genes and the Annotation of Cell Types (ACT) server[51] (Fig. 2C and Supplementary Data 1; see "Methods"). We identified PG *(IFI6, RGS5, KITLG)*, germline *(POU5F1, NANOG, ITGA6)* and stromal *(DCN, LUM, PDGFRA, NR2F2)* populations, as well as epithelial *(KRT18, KRT6)*,

mesenchymal *(VIM, STMN1)*, immune *(CD58, IL7R, TCF7, CD3G)* and smooth muscle *(ACTA2, TTN, TPM2)*. Hepatocytes *(SERPINA1, APOA1, HBA1/2, FGG)* could also be identified, reflecting fetal liver tissue captured towards the edge of the selected field of view (FOV). Similar results were obtained with W6_2 (Supplementary Fig. 2E and Supplementary Data 1). Some clusters remained undefined, likely as the 1K selective gene panel is not expansive enough to definitively resolve their identity. Cell segmentation using the assigned clusters yielded a spatial representation comparable to our initial immunofluorescence analysis. Specifically, PG cells make up the majority of the W6 ovary, with stromal cells and germline cells interspersed between the PG cells (Fig. 2C and Supplementary Fig. 2E).

Immunofluorescence analysis at W5 confirmed NR5A1 was expressed in the adrenal gland and throughout the ovary, including by the rare FOXL2+ cells (Fig. 2D and Supplementary Fig. 2F). We

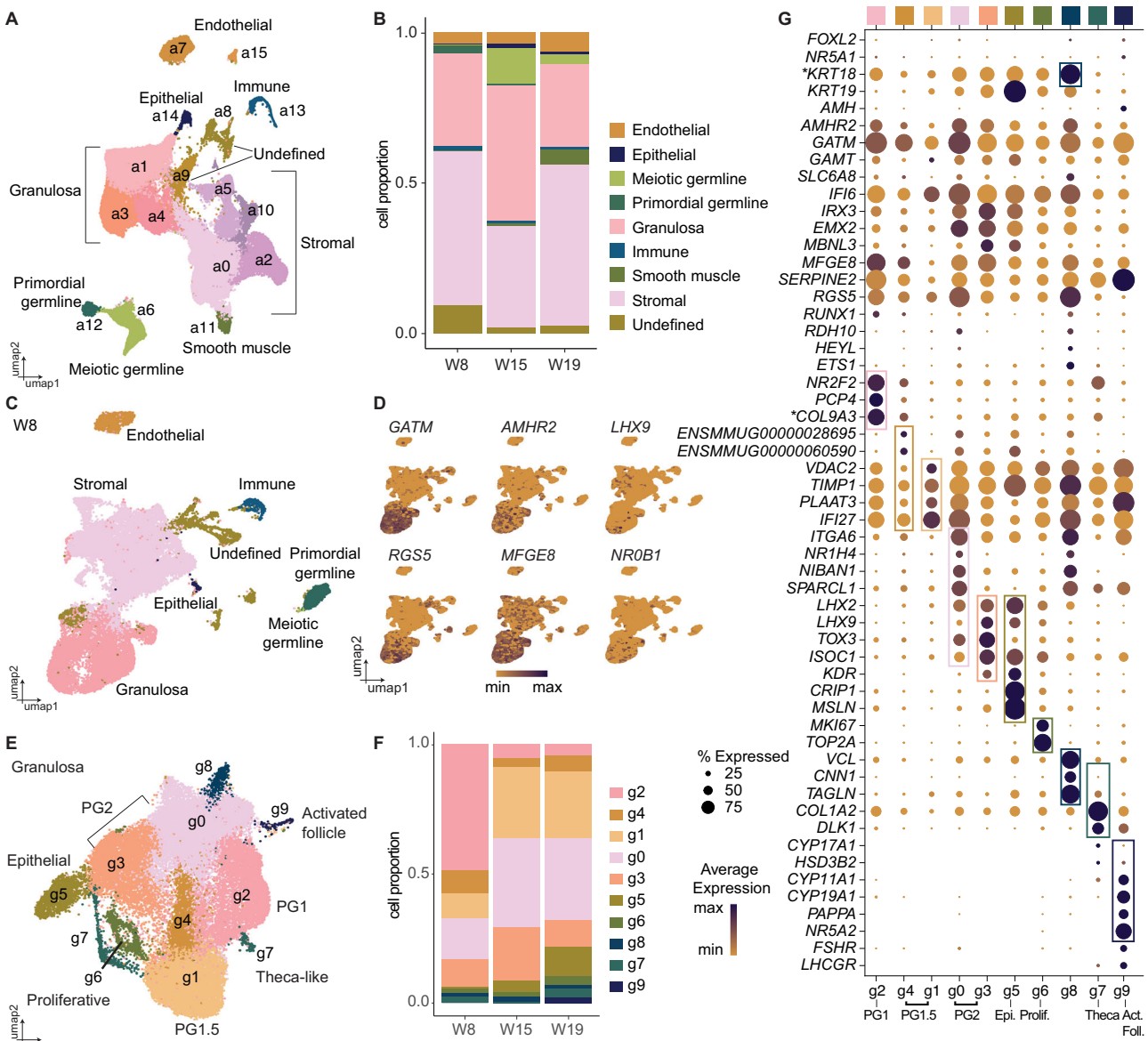

**Fig. 3 | PG cells undergo a major transcriptional shift between embryonic and fetal stages. A** UMAP plot of single cells collected from rhesus fetal ovaries at W8, W15, and W19, colored according to Seurat clusters (*n* = 3 biological replicates per timepoint). **B** Bar graph of annotated cell type proportions at each time point. **C** UMAP plot of W8 cells colored by cell annotation from the analysis in Fig. 3A. **D** Expression of ovary-enriched genes identified in the Visium CytAssist spatial transcriptomics analysis at W5 and W6 cast on the UMAP plot from Fig. 3C.

Normalized expression plotted on a high-to-low scale (indigo-yellow). **E** UMAP plot of granulosa subset (clusters a1, a3, and a4 from 3A) re-clustered and colored according to Seurat clusters. **F** Bar graph of granulosa sub-cluster proportions at each time point. **G** Dot plot of known granulosa-associated genes, or genes identified as highly enriched in each cluster (rectangles). Expression plotted on a high-to-low scale (indigo-yellow); dot size reflects percentage of cells expressing given gene. *KRT18 = ENSMMUG00000031911*), *COL9A3 = ENSMMUG00000016859.*

detected GATM in a subset of NR5A1+ cells located largely in the center of the ovary, including the NR5A1+ FOXL2+ cells (Fig. 2D and Supplementary Fig. 2F). This NR5A1+ GATM+ population positions early somatic progenitors and ESGCs (which are both NR5A1+) in the center of the developing ovary, consistent with observations in early human ovarian development[13]. WT1 and GATA4 were expressed throughout the ovary and mesonephros, while GATA2 was enriched in the adrenal gland and parts of the mesonephros, and PAX8 in mesonephric tubules (Fig. 2D and Supplementary Fig. 2G). At W6 substantial overlap was observed between NR5A1 and FOXL2, with NR5A1-bright FOXL2-dim or -absent cells closer to the ovarian cortex, and NR5A1+ FOXL2+ cells in the center with inversely correlated expression levels (i.e. NR5A1-dim likely to be FOXL2-bright and vice versa). GATM expression overlapped with FOXL2 in the center of the ovary, with additional GATM+ NR5A1+ cells present in the cortex.

Altogether, this suggests that patterning in the ovarian cords is initiated between W5 and W6, with FOXL2 protein upregulated in GATM+ progenitor cells of the WT1+ NR2F2+ NR5A1+ GATA4+ cords, most notably in the center. This co-expression pattern suggests the emergence of putative first-wave primate PG cells in the center of the developing ovary between W5 and W6 from cells with mesenchymal-like identity[14,52].

## PG transcriptome shifts between embryonic and fetal stages

To monitor ovarian cell progression, we analyzed a total of 99,247 single cells from W8, W15, and W19 rhesus ovaries using the 10× Chromium single-cell RNA-sequencing platform. We applied uniform manifold approximation and projection (UMAP) and following clustering analysis, annotated cell types based on previously published markers (Fig. 3A and Supplementary Fig. 3A). Dot plots were generated

to illustrate the percentage of cells in each cluster that expressed a given marker gene (Supplementary Fig. 3B), and individual UMAPs generated for each timepoint (Supplementary Fig. 3C). Cluster a (all) 12 corresponded to PGCs (*SOX17*, *TFAP2C*), whereas cluster a6 corresponded to meiotic germ cells (*SYCP2*) and primordial oocytes (*ZP3*). Stromal cells (clusters a0, a2, a5, a10) expressed *NR2F1, NR2F2, TCF21* and *PDGFRA*, while PG cells (clusters a1, a3, a4) expressed *KRT18, KRT19* and *GATM*. Clusters a7 and a15 expressed endothelial cell markers (*PECAM1, CLDN5, KDR, VWF*), and clusters a11, a13, and a14, corresponded to smooth muscle cells (*ACTA2, CNN1*), immune (*CD14, CD68*) and epithelial cells (*UPK3B, KRT18*), respectively. Two clusters (a8, a9) remained undefined but expressed markers of both PG and stromal cells (Supplementary Fig. 3B). The majority of undefined cells were present at W8 (Fig. 3B and Supplementary Fig. 3C); after isolating and re-clustering solely the W8 subset, undefined cells largely aligned with either PG or stromal cells (Fig. 3C). We carried out pseudobulk analysis to increase the statistical power of the DEG analyses on our biological replicates. At W8, comparing PG, stromal, and undefined cells showed undefined cells expressed PG- (*KRT18, IRX3, GSTA3*), stromal- (*LUM, NR2F1, TCF21*), and progenitor-associated genes (*GATA2, GATA3*) (Supplementary Fig. 3C and Supplementary Data 2). Consequently, these cells likely represent remnants of the bipotential precursor to PG1 that has yet to commit to either granulosa or stromal lineages.

We projected ovary-enriched DEGs from the W5 and W6 spatial data onto the W8 UMAP to localize them to specific cell types (Fig. 3D and Supplementary Fig. 4E). *GATM* and *AMHR2* were localized to W8 PG clusters, along with *RGS5*, a regulator of G-protein signaling that may have an anti-metastatic role in epithelial ovarian cancer[53]. *MFGE8, IFI6*, and *SERPINE2* mapped to PG clusters with some stromal expression, while progenitor markers *LHX9* and *NROB1* were restricted to a few cells, likely indicating downregulation as cells commit to PG or stromal identity. Conversely, W6 adrenal or mesonephros-enriched genes (*SULT1E1, INHBA, LUM, GATA2, OSR2*, and *OGN*), were enriched in stromal clusters, possibly reflecting a persisting mesenchymal identity (Supplementary Fig. 4E). We also compared expression in 10× scRNA-seq datasets previously collected from cynomolgus macaque fetal ovarian samples at W4–6[16,18] (Supplementary Fig. 4F, G; see "Methods"). Markers enriched in PG cells from the rhesus Visium data set, including *GATM AMHR2, RGS5, MFGE8*, and *SERPINE2* were also enriched in the 10× cyno datasets; this was particularly notable in cyno W6_2 (Supplementary Fig. 4G). Subsetting by cell type highlighted most genes were specifically localized to PG populations, similar to our rhesus W8 comparison (Fig. 3D).

We next extracted the PG subset (clusters a1, a3, a4) from the whole dataset for further analysis (Fig. 3E–G, Supplementary Fig. 5A, B, and Supplementary Data 3). We were able to identify rhesus PG subclusters that correlated with PG subtypes (PG1, PG2, and epithelial) identified in human fetal ovaries (W7–21)[13,14] (Fig. 3G). As in the human, PG subtypes co-exist in the fetal ovary with varying proportions over time. Cluster g (granulosa) 2 was enriched for human PG1-associated gene *PCP4*, and was abundant at W8 but depleted at W15 and W19. *RUNX1*, implicated in early mouse ovarian development, was detected in a fraction of cluster g2 cells but was otherwise absent (Fig. 3G), supporting the transient nature of *RUNX1* expression at the time of ovary formation in the embryo but not during the fetal stages. *COL9A3 (ENSMMUG00000016859)*, implicated in mouse early gonadal development but then downregulated in the ovary[54], was also enriched in cluster g2, along with stromal/progenitor marker *NR2F2*, suggesting some PG1 cells retain a partial bipotential progenitor identity. Clusters g0 and g3 expressed PG2-associated *LHX2, LHX9, NR1H4*, and *TOX3*, along with *ITGA6, SPARCL1* and the cell survival-linked apoptosis mediator *NIBAN1*. Cluster g8 and a proportion of cluster g0 cells also expressed retinol dehydrogenase (*RDH10*), *HEYL*, and *ETS1*, which are associated with rare granulosa cells in human fetal samples[13] (Fig. 3G). Cluster g8 was also enriched for actin-binding cytoskeletal proteins

*VCL, TAGLN*, and *CNN1*, which may be required for morphogenetic driven by oocyte-granulosa changes during folliculogenesis[55]. Cluster g5 expressed human epithelial granulosa-associated *KDR*, along with *CRIP1* and *MSLN*, implicated in ovarian cancer metastasis[56,57] and *ISOC1*, a serine protease implicated in DNA repair[58], also detected in PG2. We observed *KRT19* expression in a gradient from epithelial (high), PG2 to PG1 (low), analogous to the spatial patterning we observed in our immunofluorescence analysis (Fig. 1B). We also detected broad expression of *KRT18 (ENSMMUG00000031911)*, also expressed by bovine granulosa cells[25].

PG-associated genes identified at W5–6 remained associated with PG subtypes in later developmental. For example, *GATM* was expressed in an inverse gradient to *KRT19*, enriched in PG1 and PG2, but lower in epithelial, similar to FOXL2 protein patterning (Fig. 3G). The GATM partner guanidinoacetate N-methyltransferase *GATM* (required for creatine synthesis) and the creatine transporter *SLC6A8* (CRTR) were also expressed across PG clusters, which may imply a role for creatine metabolism in PG biology. *IRX3, EMX2*, and *MBNL3* were enriched in PG2 and epithelial clusters, while *AMHR2, IFI6, MFGE8, RGS5*, and *SERPINE2* were present at varying levels across all clusters. In the mouse, *IRX3* is transiently expressed in nascent PG cells within germline nests and promotes gap junction connections with the oocyte[59,60]. We noted that two additional PG clusters (g4 and g1) expressed low levels of PG1-asssociated genes alongside some PG2-enriched genes; consequently, we designated this population PG1.5 (Fig. 3G). As PG1.5 cluster proportions increase from W8 to W15/W19, while PG1 decreases, we hypothesize PG1.5 represents a PG1-like population that is transitioning from a progenitor identity, possibly into PG2. We also compared expression in pre-granulosa cells from 10× scRNA-seq datasets previously collected from cyno fetal ovarian samples at W8–17[16,20,61] (Supplementary Fig. 5C–E; see "Methods"). Epithelial, PG1- and PG2-analogous gene expression in cyno W7 and W12 samples[20] has been described[13] (labeled OSE, preGC-1, and preGC-II, respectively). We assayed these and an additional W8–16 cohort[61] and also identify these populations (clusters 2, 1, 0), which show similar expression patterns to their rhesus equivalents (Supplementary Fig. 5D, E). Cluster 3 shows similar expression to rhesus cluster g8. We did not identify a clear cyno PG1.5 population, possibly due to shared gene expression with both PG1 and PG2.

We performed further immunofluorescence analyses to determine the identity of FOXL2+ PG cells from W8 onwards (Fig. 4A). At W8, FOXL2+ PG cells in the center of the ovary mainly expressed NR2F2, marking these as predominantly PG1, with some FOXL2+/NR2F2-dim/negative cells closer to the cortex that likely correspond to the KRT19+ FOXL2+ PG2 cells we previously observed, implying putative second wave PG2 cells emerge between W6 and W8 (Fig. 1B). At W15 and W19, most FOXL2+ were NR2F2-negative and restricted to the ovarian cords; any residual FOXL2+ NR2F2+ cells were only observed outside of the cords. Similarly, NR5A1 and FOXL2 were initially co-expressed at W8, but by W15 most FOXL2+ cells were NR5A1-negative, with only a few residual FOXL2+ NR5A1+ cells remaining that were absent by W19 (Fig. 4A). In mice, NR5A1 expression decreases coincident with FOXL2 upregulation in the embryonic ovary, and FOXL2 has been shown to negatively regulate NR5A1 in mice and human granulosa cells[62,63]. GATM remained co-expressed with FOXL2 at all stages, confirming this as an early and persistent PG cell marker (Fig. 4A). We only detected proliferation-associated *MKI67, TOP2A*, and *PCNA* in a small subset of PG cells (cluster g6); immunofluorescence analysis confirmed almost no FOXL2+ cells co-expressed Ki-67 at W8 (average of 4.3%), W15 (0.9%), or W19 (0.4%) (Supplementary Fig. 5F, G). This correlates with previous studies in human fetal ovaries, where Ki-67 is predominantly expressed in germ cells, with only occasional expression in growing follicles and stromal cells[64].

Pseudobulk analysis revealed the main transcriptional distinction in PG cell identity was between W8 and later timepoints, with W15 and

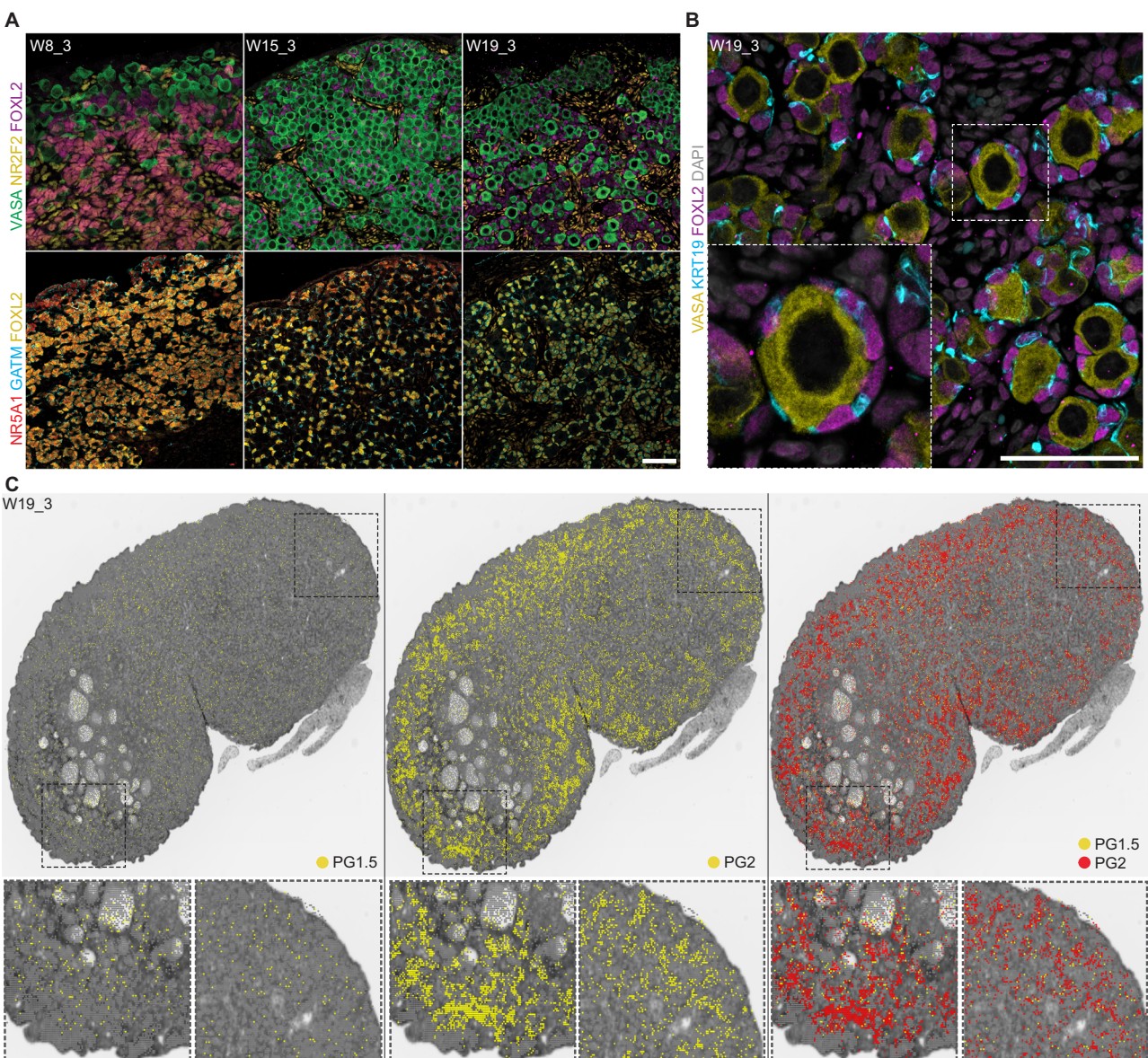

**Fig. 4 | PG1.5 and PG2 contribute to both quiescent and activated follicles.**
**A** VASA (green), NR2F2 (magenta), FOXL2 (yellow), NR5A1 (red), GATM (cyan) expression and DAPI (grey) at W8 (*n* = 3), W15 (*n* = 3), and W19 (*n* = 4). Scale bars 50 µM. **B** VASA (yellow), KRT19 (cyan), FOXL2 (magenta), expression and DAPI (grey) in primordial follicles at W19 (*n* = 4). Scale bars 50 µM. **C** Spatial distribution of germline bins (E) and granulosa PG1 subcluster bins identified in Visium CytAssist HD analysis of W19_3 mapped onto a black and white H&E image. See "Methods" for details.

W19 relatively similar to one another (Supplementary Fig. 5H, I and Supplementary Data 2). W8 enriched genes were linked to intermediate mesoderm (*FOXA2, OSR1*), indifferent gonad (*NROB1, HAO2, LGR5, RUNX1*) or stromal (*NR2F2*) identities, correlating with a persistent progenitor signature. PG1/1.5-predominant genes (*PCP4, COL9A3*) were enriched at W8 compared to PG2/epithelial genes at D100 and D130 (*KDR, LHX2, NR1H4*), reflecting the shifting balance of PG populations with time. Genes involved in mesenchymal-to-epithelial transition (*OVOL2, ELF3*) were also enriched at W8, which may facilitate the transition towards epithelial-like granulosa specification from mesenychme-originating indifferent early somatic progenitor cells[65,66]. W15 and W19 were enriched for follicle-stimulating hormone receptor (*FSHR*) and FSH-responsive survival/proliferation-associated kinase *SGK1*, suggesting increased receptiveness to pituitary hormone signaling and preparation for resumption of granulosa proliferation/activation[67]. Altogether, this highlights that broad granulosa identity is established early in embryo development, followed by only slight

changes in subtype identity following embryonic-to-fetal transition, but massive changes in the relative proportions of PG1 and PG2 with PG2 the predominant PG cell type in the ovary by W19.

## PG1.5 and PG2 contribute to quiescent follicles

Consistent with follicle formation and activation occurring prior to birth in primates, we identified cortical primordial follicles at W19, alongside activated follicle-like structures in the medulla of solely the W19_3 replicate (Supplementary Figs. 1 and 2). Activated follicles remained present at 6MPN, when mini puberty has been observed in the rhesus macaque[7]. The majority of these fetal activated follicles appeared morphologically distinct from post-pubertal follicles, with fetal follicles being more cyst-like—forming a large antrum without first establishing multiple layers of granulosa cells around the oocyte. We sought to classify the PG cells that made up the primordial follicles at W19—our immunofluorescence analysis identified these as squamous FOXL2+/KRT19+ cells (Fig. 4B), but they did not seem to have a distinct

identity in our 10× analysis. PG1 and PG2 cells initially exhibited spatial patterning, with a high-low gradient in FOXL2 expression from medulla to ovarian surface epithelium, and an inverse gradient of KRT19 expression, similar to early human ovaries[14] (Fig. 1B). However, by W15, we found all FOXL2+ PG cells in the rhesus ovarian cords were KRT19+. Consequently, we sought to investigate the contribution of PG1, PG1.5, and PG2 populations to the granulosa cells in either the quiescent or activated follicle types at W19.

To do this, we performed spatial transcriptomics on W19_3, using the 10× Genomics Visium HD CytAssist platform with the human whole transcriptome probe set, and used the clusters identified in our 10× Chromium analysis to assign cell type identity to given bins (Fig. 4C; see "Methods"). We first interrogated the spatial distribution of the germline at W19 as a positive control. As in previous human analysis, we identified multiple germline stages in our 10× Chromium when we further analyzed our germline subset (clusters a6 and a12, Fig. 3A); primordial germ cells (PGCs, clusters gL2, gL5), meiotic germ cells (MGCs, gL0), retinoic acid-responsive MGCs (gL3, gL1), and primordial oocytes (gL4)[17] (Supplementary Fig. 6A, B). PGCs made up the majority of the W8 germline, but greatly reduced by W15 and W19 as the germline differentiated into later stages (Supplementary Fig. 6C, D). In line with this, primordial germ cell identity was rarely detected in our spatial dataset (Supplementary Fig. 6E). Primordial oocytes were distributed throughout the ovary, while retinoic acid (RA)-responsive meiotic germ cell bins were largely arranged in the outer cortex. Turning to our PG clusters, we only identified rare PG1 identity in the ovary at W19, consistent with our 10× Chromium data (Supplementary Fig. 6F). These PG1-assigned bins were randomly distributed throughout the ovary. PG2-assigned bins were detected in both the cortical primordial follicles and in the medullary activated follicles, as were PG1.5-assigned bins (Fig. 4C). Therefore, although PG2 cells were the predominant PG type by W19, they were not preferentially associated with a given follicular identity or ovarian region. Instead, we found primate quiescent and activated follicles were made up of a mix of PG cells. This suggests that, similar to mouse, human, and cyno, PG1 cells initially found in the medulla of the developing ovary between W5 and 6 are not major contributors to quiescent follicle establishment in the rhesus ovarian reserve. We hypothesize that the loss of spatial patterning in later gestation (W15 or earlier) may be due to a fate transition from PG1 to PG2 through a PG1.5 intermediate and/or the rapid expansion of the ovarian cords observed in both human and macaques (distinct from the mouse), which may reshuffle somatic cells as germline cells proliferate. Consequently, we posit the establishment of quiescent follicles is instead related to spatial position in the ovary instead of PG type, although functional models will be needed to address this.

## Early activated follicles show evidence of hormone production at W19

Increased ovarian sex-steroid secretion has been noted after W16 in human ovaries[68], and explanted tissue shows increased capacity for in vitro steroid conversion at W17−21[69]. The enzymes responsible for sex-hormone production prior to birth have previously been identified at W17−20 using bulk organ qPCR for *CYP17A1*, *CYP19A1*, *HDS3B2*, *CYP11A1*, and estrogen receptor genes (*ESR1*, *ESR2*) with protein staining in somatic cells[70]. Similarly, hormonal measurements showed rhesus fetal ovaries are competent to produce estrogen (via estradiol) between W15 and W19, leading to speculation both theca and granulosa cells would be required[71]. We detected a W19-specific granulosa subtype (cluster g9) in our 10× dataset that expressed genes associated with activated granulosa cells (Fig. 3G and Supplementary Data 3). These included steroidogenic enzyme-encoding *CYP19A1*, *CYP11A1*, and *HSD3B2* (3βHSD2), pregnancy-associated plasma protein-A (*PAPPA*), inhibin b (*INHBB*), and *NR5A2*, which has been shown to interact with Gata4 in mice to promote 3βHSD2 and CYP19A1

expression[72]. Cluster g9 was also enriched for *FSHR* and LH receptor (*LHCGR*), as well as *AMH*, also detected in activated follicles in human fetal studies and one of the hormones produced during mini puberty[5,64,73] (Fig. 3G). We also noted that cluster g7 was enriched for theca-associated genes (*COL1A2, CYP17A1, DLK1, HSD3B2*), while also expressing PG-associated genes *GATM, IFI6*, and *SERPINE2*, and the stromal marker *NR2F2* (Fig. 3G). Cluster g7 was divided on the PG UMAP; only half expressed *CYP17 A1* and *HSD3B2*, while *DLK1* and *COL1A2* were detected in both (Fig. 3E and Supplementary Fig. 7A).

To characterize these putative fetal theca cells, we extracted and re-clustered cluster g7, identifying three clusters (Supplementary Fig. 7B, C and Supplementary Data 3). One cluster, t1 (theca1), was enriched at W8, while the other two (t0, t2) were enriched at W19 (Supplementary Fig. 7B, C). We examined the expression of mouse or human theca-associated genes identified from the literature[74–79] (Supplementary Fig. 7D). In mouse ovaries, lineage tracing identified steroidogenic *Gli1+ Cyp17a1+* mesonephros-derived theca that migrate into the ovary and *Wt1+ Esr1+* (estrogen receptor) ovarian stromal-derived theca cells that later acquire *Gli1+* expression[78], as well as CYP17A1+ theca cells differentiated from Foxl2+ granulosa lineages, Theca-G, and Foxl2− CYP17A1− stromal derived cells, Theca-S[79]. Adult human theca is thought to be stroma-derived, with structural, androgenic, and perifollicular subtypes[78].

Cluster t1, predominantly found at W8, lacked steroidogenic/androgenic genes but expressed *GATM, NR2F2*, and *TCF21*, theca progenitor-associated genes and genes associated with W8 granulosa (*PNLIPRP2, HAO2*) and indifferent gonadal progenitors (*RSPO1, NROB1*) (Supplementary Fig. 7D, Supplementary Fig. 5E, G, and Supplementary Data 3). Cluster t2 expressed steroidogenic, theca progenitor- and theca interna-associated genes as well as *LHCGR*. (Supplementary Fig. 7D and Supplementary Data 3). Cluster t0 expressed PG- and structural theca-associated genes, with minimal expression of steroidogenic and progenitor markers. Cluster t0 was also enriched for *THY1*, implicated in theca cell layer adhesion in mouse follicles[80], and expressed proliferation-associated genes *MK167* and *TOP2A* (Supplementary Fig. 7D). Mouse Theca-G associated genes (*HIF1A, CREB3L2*) could be detected in the three theca clusters, mainly enriched in cluster t2, while only Theca-S associated *CREM* was appreciably but lowly detected (Supplementary Fig. 7D). *WT1* or *GLI1* were not appreciably expressed in the theca clusters. By contrast, steroidogenic genes were not appreciably detected in our stromal cell clusters and additional theca-associated genes were broadly but lowly expressed (Supplementary Fig. 7E; cluster subset a0, a2, a10, and a5 from Fig. 3A). Given the residual expression of granulosa genes, we speculate that the fetal theca we observe may originate from or share a common progenitor with PG lineages, similar to Foxl2-derived mouse Theca-G cells (albeit with some distinctions as mouse Theca-G cells are CYP17A1-negative)[79]. However, the precise origin of these cells will require validation in future studies. We also find estrogen receptor (*ESR1*) and androgen receptor (*AR*) genes enriched in granulosa, stroma, smooth muscle, and undefined cells predominantly at W19, suggesting the fetal ovary may be receptive to hormonal signaling (Supplementary Fig. 7F).

We noted that cluster g9 was made up solely of cells from the W19_3 replicate, where we had observed activated follicles in the medulla (Supplementary Fig. 2). Spatial transcriptomics and immunofluorescence analysis localized the activated granulosa cells and putative fetal theca subtypes to these fetal activated follicles (Fig. 5A). Bins with a *CYP19A1*+ activated granulosa identity (cluster g9) were detected in the large activated cyst-like follicles in the medulla, surrounded by *CYP17A1*+ theca cluster t2 (Fig. 5A). Immunofluorescence analysis showed CYP17A1+ cells surrounding the W19_3 and 6MPN follicle-like structures (Fig. 5B). FOXL2+ granulosa within the medullary activated follicles also expressed PAPP-A or CYP19A1, or both (Fig. 5C, insets). A subset also expressed AMH, which we also detected

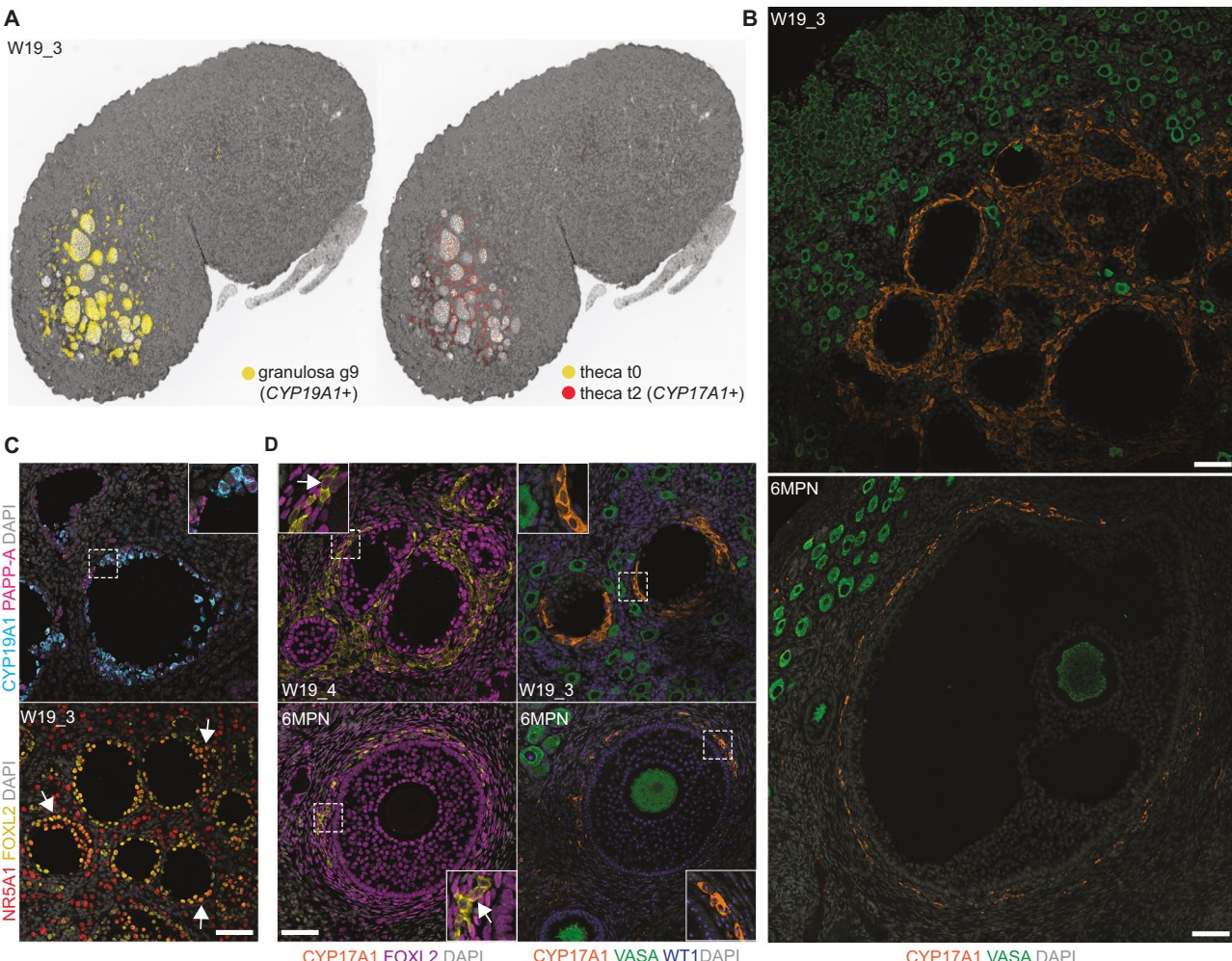

**Fig. 5 | Early activated follicles recruit fetal theca cells and produce hormones.**
**A** Visium CytAssist HD spatial distribution of granulosa cluster g9 and theca clusters t0 and t2 bins identified in W19_3 mapped onto a black and white H&E image.
**B** VASA (green), CYP17A1 (yellow) expression, and DAPI (grey) at W19 ($n = 4$) and 6 MPN ($n = 1$). **C** NR5A1 (yellow), FOXL2 (magenta), PAPP-A (red), CYP19A1 (cyan) expression and DAPI (grey) in fetal activated follicles at W19 ($n = 4$). Magnified area (dashed) is shown in the inset. **D** VASA (green), CYP17A1 (yellow), FOXL2 (magenta), WT1 (blue) expression and DAPI (grey) at W19 and 6MPN. Magnified area (dashed) is shown in the inset. All scale bars 50 μM.

in W8 ovaries, but not at W15 (Supplementary Fig. 7G). Previous studies only noted AMH expression in human fetal ovaries after 36 weeks[64,73], but as the earliest samples in these studies were collected after the equivalent of our rhesus W8 timepoint, we suspect that AMH is transiently expressed in PG1. FOXL2+ NR5A1+ cells were also observed near the activated follicle-like structures, as well as in smaller clusters in the adjacent medulla D130 and 6 MPN (Fig. 5B and Supplementary Fig. 7G, arrows). These stromal NR5A1+ cells are consistent with androgenic theca[78]; NR5A1 expression has been observed in both granulosa and theca cells in human ovarian follicles after the preantral stage[81], and in mouse follicular granulosa cells[82].

We did not detect CYP17A1+ cells in W19_1 and W19_2, or earlier at W8 or W15. We therefore included two additional D130 samples; W19_4, where we observed follicles similar to W19_3 (Fig. 5D and Supplementary Fig. 7H), and W19_5, which instead resembled W19_1 and W19_2 (no large follicle-like structures or CYP17A1-expressing cells). This suggests rhesus fetal activated follicles emerge around W19, approximately 35 days before birth. We also did not identify any active follicle granulosa or fetal theca gene expression in the cyno dataset (Supplementary Fig. 5C), which is mainly composed of earlier developmental samples. However, active follicles in the cyno have also only been observed around 2.5 MPN[15], distinct from rhesus and human[83] where they emerge in late gestation. In human ovaries,

CYP17A1 was only detected briefly between W19 and 24 and then after W33—this latter timepoint correlates with our rhesus W19[83,84]. Despite retaining PG transcripts, CYP17A1+ cells were predominantly FOXL2-negative (Fig. 5D; arrows show rare possible FOXL2-dim double-positive cells in insets). CYP17A1+ cells were also largely WT1-negative, in contrast to WT1+ neighboring granulosa cells (Fig. 5D, insets). WT1 dysregulation in mouse ovaries results in upregulation of NR5A1 and steroidogenic genes in granulosa cells; perhaps a similar downregulation is occurring here as these fetal cells transition towards a theca fate[85,86].

As the early activated follicles are eventually lost, we also examined the expression of apoptosis-associated markers cleaved Caspase3 (cCasp3) and cleaved poly ADP-ribose polymerase (cPARP) by immunofluorescence at W19 and 6 MPN (Supplementary Fig. 8). At W19 we did not observe any cCasp3 or cPARP staining in FOXL2+ or CYP17A1+ cells, and only very rare additional somatic staining (Supplementary Fig. 8A). However, by 6 MPN, we detected cCasp3 in granulosa cells in early activated follicles (Supplementary Fig. 8B). This suggests these structures are beginning to degrade, signaling the decline of the fetal endocrine period and the end of minipuberty. cCasp3 staining was also higher in oocytes in medullary activated follicles relative to cortical primordial follicles (Supplementary Fig. 8C, D). We did not detect cPARP staining in the fetal activated follicles (observing only rare

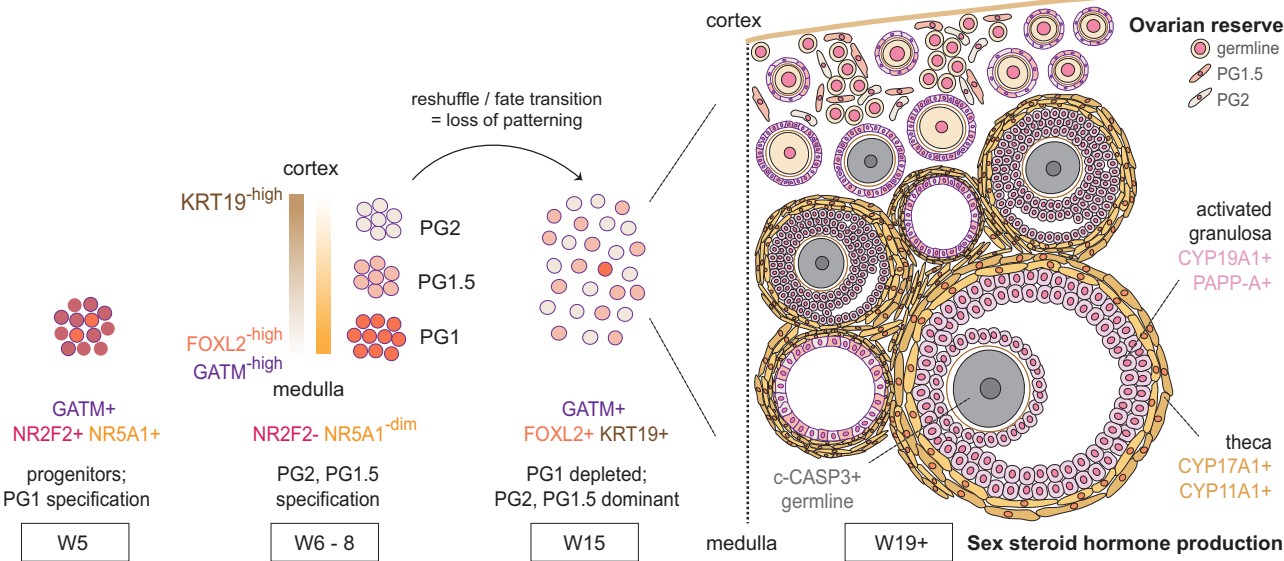

**Fig. 6 | Model of primordial and early activated follicle formation in primate gestation.** As in the human, rhesus prenatal ovaries develop multiple types of pre-granulosa (PG) cells. PG1 cells are specified first at W5, with PG2 and PG1.5 emerging between W6 and W8. PG2 and PG1.5 contribute to primordial and early activated follicles, while PG1 is largely depleted by late gestation. Medullary follicles composed of activated granulosa cells also recruit fetal theca cells, establishing a two-cell system for sex-steroid hormone production prior to birth.

bright staining in oogonia within the remaining nests), suggesting that oocyte cell death in early activated follicles is mediated primarily via cleaved Caspase3. In human fetal ovaries, rare cCasp3 expression has been described from W16−41 in oocytes and follicular granulosa cells, though the spatial location of the follicles was not noted[87]. Granulosa cells in the human fetal ovary generally showed weak staining for apoptosis factors (Bok, cCasp3, Tnf) until they were located in secondary or antral follicles[87,88]. Overall, our data supports the hypothesis that a simple two-cell circuit of androgen-producing fetal theca and aromatase-producing fetal granulosa cells in activated follicles develops before birth. We propose that it is required for early hormone production, which switches from the placenta to the new-born after birth resulting in estrogen production during mini-puberty[89–91].

## Discussion

This work represents a singular study of rhesus macaque ovarian biology over the entire gestational time course, especially during two developmental windows that are almost inaccessible with human tissues: gonadal ridge expansion and ovarian reserve formation (see model in Fig. 6). We identify a gonadal signature in W5–6 ovaries that is distinct from the emerging adrenal gland and mesonephros, with first-wave PG1 cells emerging from indifferent mesenchymal progenitors in the center of the ovary, likely from the early somatic progenitors and ESGCs observed at W5. We show that the two major PG populations in humans, PG1 and PG2, are identified in rhesus macaque by W8, indicating that a second wave of PG cells emerges between W6 and W8. We also identified a third major PG population at W8 with shared characteristics between PG1 and PG2 that we called PG1.5. It is these later PG cells (PG1.5 and especially PG2) that ultimately become the dominant PG cell type in the ovary at the time of nest breakdown and follicle formation. Genes such as GATM and IFI6 are early markers of PG cells that persist throughout development; further characterizing the function of these and other lineage-associated genes will likely improve our understanding of differences of sex development or developmental origins of ovarian disease and dysfunction.

In agreement with data from human ovarian development[13], we identify initial spatial patterning of PG1 and PG2 granulosa cell sub-types, with the W6 ovary composed of PG1 cells in the center of the ovary and PG2 cells close to the cortex. At W8, there are also clearly two major types of granulosa cells that are spatially organized, and the character of these cells is distinctly epithelial-like (PG2) and mesenchymal-like (PG1). In addition, consistent with second wave preGC-II/PG2 cells in humans[13,14], LGR5 is also not expressed by second wave PG1.5 or PG2 cells in the rhesus ovary. This is distinct from the mouse, where LGR5 is a major marker of second wave PG cells[12]. Data in the rhesus indicate that a major transcriptional shift in PG cell identity occurs after the embryonic-fetal transition (W8), where the PG population switches from mostly PG1 to mostly PG1.5/PG2 at W15 and W19. Using spatial transcriptomics at the time of nest breakdown, follicle formation, and follicle activation (W19), we show that PG2 and PG1.5 contribute to both the granulosa cells of the primordial follicles in the cortex and the granulosa cells of activated follicles in the medulla. In contrast, PG1 is very rare and, when detected spatially, is randomly allocated to the different types of follicles. Given this, granulosa derivation protocols for in vitro follicle generation could be simplified by focusing on the generation of second wave PG cells. Altogether, our data sheds new light on the fate of PG cells during the establishment of different types of ovarian follicles that have not been documented in any primate species, including humans.

As nest breakdown occurs, the earliest follicles to develop immediately activate, leading to specialized activated granulosa cells that recruit fetal theca cells. Both activated granulosa and theca cells in fetal life share gene expression patterns with their later developmental cognates; however, fetal follicles appear mostly cyst-like, suggesting that fetal follicles are not equivalent to reproductive follicles. We also find the germline cells in these follicles are preferentially fated for caspase-3 mediated apoptosis. Previous human and rhesus studies describe an increase in androgen/estrogen production by the ovary in later gestation; our rhesus data links this to (1) the W19 emergence of granulosa/theca cells in activated follicles, and (2) locates these hormonal units to the cortico-medullary region of the ovary. Data in the later stages of human ovarian development is rare, but the morphological, cellular, and molecular similarities with the rhesus ovary suggest that these conclusions are likely applicable to humans as well. We propose that this provides an explanation for fetal endocrine activity linked to mini puberty. It remains unclear why only medullary follicles gain endocrine capacity, while those in the cortex are shielded. Determining what triggers this activation could have implications for

understanding polycystic ovary syndrome (PCOS), which is also characterized by growth of multiple immature follicles. Similarly, it remains unclear what drives the quiescence of the HPG axis after this transient activation.

Although this work expands the body of data available on primate ovarian development, especially using emerging technologies, several unanswered questions remain that we hope will be addressed in future studies. Overall, the human ovary remains an understudied organ, despite its critical role in reproduction and hormonal production. Our understanding of ovarian dysfunction, such as in PCOS or primary ovarian insufficiency (POI), has been hampered by this knowledge deficit, as has generating human ovarian models in vitro from stem cells. We consider the rhesus macaque an important complement to the excellent work of colleagues in the early cynomolgus macaque and human. Altogether, this work creates a comprehensive data set revealing the spatial origins of the primate ovary, tracing the fate and spatial patterning of ovarian cells through the major ovarian remodeling and ultimately the establishment of primordial follicles from PG cells.

## Methods

### Rhesus macaque sample collection
The research in this manuscript complies with all relevant ethical regulations, including the ONPRC Institutional Animal Care and Use Committee (IACUC) and the UCLA Chancellors Animal Research Committee (ARC) as described below.

All animals were housed at the ONPRC, an American Association for Accreditation of Laboratory Animal Care accredited institution. Animals are cared for in accordance with the Guide for the Care and Use of Laboratory Animals. Animal procedures were performed by trained veterinary and technical staff in accordance with the Public Health Services Policy on Humane Care and Use of Laboratory Animals. Rhesus macaque time-mated breeding experiments were conducted following the approval of the ONPRC IACUC with secondary approval by the UCLA Chancellors ARC for all experiments with rhesus tissues performed at UCLA.

Time-mated breeding was performed by measuring female estradiol levels daily from day 5-8 after menses began, then pairing with a fertile male once estradiol readings had risen above baseline (> 50 pg/mL)[92]. The male was removed twenty-four hours after ovulation, as measured by the estradiol peak[93]. Day 1 (D1) of embryo development was estimated to occur 72 h after the estradiol peak. Pregnancy was confirmed by measuring progesterone levels and by ultrasound. Fetal sex was determined using an adaptation of a previously described assay[94]. The Y-chromosome specific genes DYS14 and SRY were PCR amplified from rhesus macaque genomic DNA (gDNA), purified, cloned into a pGEM-T Vector, and transformed into JM109 competent cells (pGEM-T Easy Vector Systems, Promega). Conventional PCR was performed to identify clones with the correct insert, and subsequent plasmid DNA was extracted and purified using the GeneJET Plasmid Miniprep Kit (Thermo Fisher Scientific). Standard curves were generated using purified plasmid DNA and used to determine DYS14 and SRY levels. To determine sex at W5–6, a small piece of fetal tissue collected at the time of necropsy was used for gDNA isolation using the GeneJET Genomic DNA Purification Kit (Thermo Fisher Scientific) according to the manufacturer's instructions. To determine sex at W8–19, maternal blood was collected to obtain circulating cell-free DNA (ccfDNA). Blood samples (4 ml collected into a no-additive Vacutainer tube) were centrifuged, and the resultant serum was immediately used to isolate ccfDNA using the QIAmp MinElute ccfDNA Kit (Qiagen) following the manufacturer's instructions. DYS14 and SRY presence were determined by qPCR in the extracted gDNA and ccfDNA samples. Sex was determined to be female if DYS14 and SRY were not detected, whereas sex was determined to be male if DYS14 and SRY were detected at a level above the minimum plasmid standard.

Time-mated offspring were collected by C-section ($n = 4$ at W5 (D34, Carnegie Stage (CS) 16; two XX, two XY), $n = 3$ at W6 (CS20, D41; two XX, one XY), $n = 3$ at W8 (CS23, D50-52), $n = 3$ at W15 (D100), $n = 3$ at W19 (D130); W8–19 timepoints all XX). At W5–6, whole embryo torsos were collected following C-section and embedded; from W8 onwards, gonads were dissected from embryos. One gonad was shipped to UCLA in Hanks balanced salt solution media for further processing, while the other was fixed at ONPRC in 4% paraformaldehyde (Fisher Scientific) overnight. Tissues were then dehydrated in a series of ethanol solutions (70, 80, 95, and 100%) and xylene for subsequent paraffin embedding; ovaries were embedded longitudinally, and torsos were embedded transversely for serial sectioning (5 μm) at Oregon Health & Science University (OHSU) Histopathology Shared Resource. Every tenth sectioned slide was stained for hematoxylin and eosin (H&E). Additional slides were kindly provided from tissue samples previously collected in house at ONPRC ($n = 2$ at W19; $n = 1$ at 6 months postnatal, 6 MPN; all XX).

### Rhesus gonadal dissociation
Gonads were rinsed twice in PBS, then placed into dissociation media (10% collagenase IV, 2.5% dispase, 10% FBS, 0.1% DNase I in 1 ml PBS). Gonads were incubated at 37 °C for 15–30 min (age dependent), triturating with a P1000 pipette every 5 min to break down the tissue into a single cell suspension. 5 ml of mouse embryonic fibroblast media was added to quench enzyme activity and the resulting cell suspension centrifuged for 5 min at 1600 rpm. The supernatant was discarded and the cell pellet resuspended in 0.04% BSA in PBS. Cell number and viability were determined using a hemocytometer and Trypan Blue staining. 40,000 cells per replicate (two replicates per sample) were submitted for 10x Genomics analysis, targeting 10,000 cells.

### Immunostaining
Sections were deparaffinized and rehydrated using xylene followed by a graded ethanol series (100%, 95%, 70%, 50% ethanol), water and PBS (2 × 5 min washes). Antigen retrieval was performed in a hot water bath (95 °C) for 40 min using Tris-EDTA (pH 9.0) or sodium citrate (pH 6.0) solution (antibody dependent; see Supplementary Table 3). Slides were washed in PBS, 0.2% Tween-20 (PBS-T) and subsequently permeabilized in PBS, 0.05% Triton X-100 for 10 min. Slides blocked with 10% normal donkey serum for 30 min, then incubated with primary antibodies diluted in PBS-T (1:100–1:200) overnight at 4 °C. Sections were rinsed thrice in PBS-T before incubation with secondary antibodies diluted in PBS-T (1:400) for one hour at room temperature. All antibody details provided in Supplementary Tables 3 and 4. Slides were treated with TrueBlack® Lipofuscin Autofluorescence Quencher (Biotium 23007) per manufacturers protocol, then stained with DAPI for ten minutes before sealing with Prolong Gold antifade mountant (Invitrogen P10144).

### Microscopy and image analysis
H&E images were taken on an Olympus BX-61 light microscope, and tiles were combined where required using the Stitching plugin available in Fiji microscopy image analysis software[95,96]. Immunostained samples were imaged on an LSM 880 (Carl Zeiss) confocal microscope controlled by Zen Black software with Plan-Apochromat 10×/0.45 NA, 20×/0.8 NA or oil immersion 40×/1.4 NA M27 objectives at room temperature. Acquired images were processed using Fiji[96]. Cell counts were performed in Fiji using Cell Counter. Graphs and statistical analysis (two-way ANOVA with Tukey's multiple comparisons test) were prepared in Prism 7 (GraphPad).

### 10× Visium CytAssist spatial transcriptomics (ST) library preparation
Slides were prepared for spatial transcriptomics using the manufacturer's Demonstrated Protocol (CG000520 Rev. B, 10× Genomics).

Briefly, slides were deparaffinized as detailed for immunofluorescence analysis above. Slide RNA quality was assayed on sections removed from an adjacent FFPE slide prepared at the same time as the target spatial slide. RNA for quality assessment was extracted using an RNEasy FFPE kit (73504, Qiagen) and DV200 scores measured on an Agilent 2100 Bioanalyzer (Agilent) with the RNA Pico 6000 kit (5067–1513, Agilent). Sample scores were >30%, as per 10× Genomics recommendations. Spatial slides were H&E stained and imaged on a Keyence BZ-X series microscope at 20× magnification with tiling to capture the entire capture area, then de-crosslinked. Libraries were constructed according to Demonstrated Protocol–CG000495 using the Visium Human Transcriptome Probe Kit (V2, PN-1000466, 10× Genomics), which has 3–5 probes designed per target gene. Briefly, probes were hybridized to the sample's RNA, then ligated. Ligation products were released from the tissue and captured on the Visium slides with an additional UMI barcode. Libraries were then amplified and pooled. Sequencing was completed on a NextSeq 2000 instrument at a minimum of 25,000 read pairs per tissue-covered spot on the Capture Area, with samples multiplexed to maximize the run of data.

## 10× Visium CytAssist ST analysis, data alignment and pre-processing

Libraries were demultiplexed using the Chan Zuckerberg Biohub and fastq files aligned using spaceranger-2.1.0 software (10× Genomics). The default reference genome option was used for alignment ("–transcriptome", refdata-gex-GRCh38-2020-A) as probe sets were designed against the human genome. For the probe set list option ("–probe-set"), a rhesus macaque (*Macaca mulatta*)-specific probe subset was extracted (Supplementary Fig. 3b). The rhesus macaque has ~93% genomic similarity to the human genome and in silico prediction determined that for ~67% of the human genes, all target probes were predicted to detect the *M.mulatta* gene orthologue (single hit). For a further ~15% of genes, some probes were predicted to hit the *M. mulatta* gene ortholog (mixed multi-probe–single). The remaining probes either detected multiple *M. mulatta* genes (mixed multi-probe –multiple, multiple) or did not detect any genes (no hit). Using code provided by 10× Genomics, the single hit and mixed multiprobe–single hit probe sets were extracted and combined to generate a modified *M. mulatta* target probe set. To create the Loupe Alignment Files ("–loupe-alignment"), the CytAssist slide images and the Keyence BZ-X H&E images were imported into LoupeBrowser v7.0.1, manually aligned, and the dots overlaying tissue manually annotated as containing tissue. Data were run on a High-Performance Computing Cluster at UCSF, C4, using an array-based script. Library and mapping statistics can be found in Supplementary Table 2.

## 10× Visium CytAssist ST differential gene expression (DEG) analysis

Count matrices and associated tissue images were analyzed using Seurat (v4.3.0) in R (v4.2.2)[97]. For each sample, Visium spots with less than 100 UMI counts (nCount_Spatial) and less than 100 genes (nFeature_Spatial) were filtered out. The remaining spots were normalized using the SCTransform function (with assay = "Spatial" and vst.flavor = "v2", among other default parameters), followed by Principal Component Analysis (PCA). Default parameters were utilized for FindNeighbors, FindClusters, and RunUMAP functions. The Linked-DimPlot Seurat package was employed for manual selection of spots corresponding to gonads, mesonephros and adrenal glands, guided by prior manual spot annotation. Differentially expressed genes (DEGs) between gonads and other tissue sections were identified using the FindMarkers function, with significance thresholds set as FDR < 0.05 and log2FC > 1 or <−1).

For comparison between Visium datasets, samples were integrated using Seurat functions (SelectIntegrationFeatures (selecting nfeatures = 3000), PrepSCTIntegration, FindIntegrationAnchors (selecting normalization.method = "SCT"), and IntegrateData (selecting normalization.method = "SCT")). PCA was performed and the principal components explaining more than 90% of the data variance were used for UMAP analysis. To identify gonadal DEGs between samples, PrepSCTFindMarkers was used followed by the FindMarkers function, with significance thresholds set as FDR < 0.05 and log2FC > 1 or <−1). Bar graphs were made with Prism 7 (GraphPad).

## Nanostring CosMx ST library preparation and gene expression analysis

Samples were prepared for analysis on the CosMx platform following established protocols for tissue preservation, sectioning, and staining. Slides were deparaffinized and rehydrated using xylene followed by a graded ethanol series preparation, water, and PBS, then digested with Proteinase K followed by probe hybridization using the 1000 gene (1K) Human Universal Panel. Slides were then stained with DAPI (nucleus), B2M (cellular segmentation), PanCK (epithelial), and CD45 (immune) markers. 30 FOVs were selected (13 in W6_3, 17 in W6_2), imaged, and collected to cover ovarian, mesonephric, and metanephric regions in both embryo sections (plus adrenal in W6_3. Data acquisition on the CosMx platform was processed using AtoMx.

For W6_3, the cell segmentation results for CosMx data provided by Nanostring were uploaded into Seurat using LoadNanostring function. CosMX includes Negative control probes that are modeled after synthetic sequences. Negative control probes serve as non-target controls for quantification of non-specific ISH probe hybridization[98]. Cells with more than 25 percent of negative probes, system control probes, and expressing less than 20 genes (nFeature_Nanostring) were filtered out. Transcripts containing "NegPrb" and "SysPrb" were removed, resulting in 1000 unique genes, 27344 cells, and 13 FOVs. Sample normalization was performed using the SCTransform function (selecting assay = "Nanostring", vst.flavor = "v2", clip.range = c(−10, 10) and other parameters as default) followed by PCA, clustering, and UMAP visualization. DEGs between clusters were identified using the FindMarkers function (with only.pos = TRUE, logfc.threshold = 0 parameters). Cluster annotation was based on a predefined list of canonical markers and the results from the ACT server[51]. The CosMX object was subset into individual gonads based on FOVs, and ImageFeaturePlot and Image-DimPlot functions employed for further analysis and visualization.

For W6_2, low-quality cells were filtered out using the same parameters used for Embryo 1 ("NegPrb" and "SysPrb"), resulting in 1000 unique genes, 20312 cells, and 17 FOVs. Sample normalization, PCA, FindNeighbors, FindClusters, and RunUMAP functions were performed as above, and the CosMx object subset into individual gonads. After subset, sample normalization was redone on each gonad using SCTransform function and PCA performed (using npcs = 100 and other parameters with default settings) followed by UMAP analysis. The FindClusters function was used for the cluster identification (using resolution = 3 and 2 for gonad 1 and gonad 2, respectively). For the identification of DEGs between clusters, the FindMarkers function was used (with only.pos = TRUE, logfc.threshold = 0 parameters). Cluster annotation was based on a predefined list of canonical markers, and ImageFeaturePlot and ImageDimPlot functions employed for further analysis and visualization. Library and mapping statistics can be found in Supplementary Table 2.

## 10x Chromium single cell RNA-sequencing (scRNA-seq) library preparation and analysis

scRNA-seq libraries were generated using the 10× Genomics Chromium instrument and Chromium Single Cell 3′ Reagent Kit v2. Each individual library was designed to target 10,000 cells. Libraries were generated according to the manufacturer's instructions and library fragment size distribution was determined using a BioAnalyzer instrument. Libraries were pooled together and sequenced using an Illumina Novaseq 6000 platform, at an average depth of 200 million reads per sample.

Raw GEX libraries were mapped to the Ensemble rhesus macaque reference genome (Macaca_mulatta.Mmul_10 release 109) using cellranger count (version 6.1.1) from 10× Genomics. Single-cell expression matrices generated with Cell Ranger (10× Genomics) using cellular barcodes and unique molecular identifiers (UMIs) were used for single-cell transcriptome analysis. Quality assessment and quality filtering of UMIs and individual cells was performed using the CellMembrane R package (https://github.com/bimberlabinternal/CellMembrane) with experiment specific thresholds. Mitochondrial content QC was carried out using a custom gene set (*ATP6, ATP8, COX1, COX2, COX3, CYTB, ND1, ND2, ND3, ND4, ND4L, ND5, ND6*) and cells expressing more than 15% of mitochondrial genes were filtered out. Doublets were excluded using the scDblFinder package[99]. A total of 99,247 single cells passed the QC thresholds, with an average of 1795 genes per cell; one technical replicate for W19_3 was excluded, as it did not pass the quality assessment. Data set normalization, reciprocal PCA based integration, clustering, and cluster marker identification were performed using the Seurat R package[97] (v4.4.0) with default settings. Clusters identified by the FindAllMarkers function (test = MAST[100]) were annotated as different cell types based on known markers. Barplots were generated using dittoseq[101]. UMAP plots and barplots were colored using viridislite[102] or batlow[103] scientific color map palettes. Library and mapping statistics can be found in Supplementary Table 2.

Pseudo-bulk differential expression (DE) analysis was performed in all granulosa cells and stromal cells to identify genes that are differentially expressed between time points; and in W8 granulosa, stromal, and undefined cells to identify genes that are differentially expressed between these cell types. Cells were extracted from the main dataset using the subset function in Seurat and counts were summed from all cells in each cell type for each biological replicate. DE analysis between biological conditions or cell types was performed using bulk RNA-seq DE methods, where derived per sample counts were normalized using the trimmed mean of M-values method (TMM)[104], and transformed to log-counts per million with associated observational precision weights using the voom method[105]. The undefined cells from W8_1 were excluded from further analysis due to low overall counts and high variation compared to the other replicates. Gene-wise linear models were employed using limma with empirical Bayes moderation[106] and false discovery rate (FDR) adjustment[107]. For DE between time points, models contained independent variable time point. For DE between granulosa, stromal, and undefined cell types at W8, models contained cell type and accounting for within subject correlation. For k-means clustering, differentially expressed genes (DEGs) were further filtered using FDR < 0.05 in at least one comparison and top 500 most variable genes across all comparisons. Top 10 most up or downregulated genes were determined after ranking based on log2foldchange > 1 or < −1. The R package pheatmap was used for heatmap generation and K-means clustering.

For cynomolgus macaque analyses, the following datasets were assayed – GSE160043[18] (E24/W4_1, E28/W4_2, E31/W5_1, E37/W6_1); GSE194264[61] (W8_1, W8_2, W10, W12_1, W16_1, W16_2, W18); GSE149629[20] (E84/W12_2, E116/W17) or Zenodo 6918355[16] (E35/W5_2, E40/W6_2, E52/W8_3). Sample count matrices were downloaded from their respective GEO or Zenodo repositories into Seurat (v4.4.0)[97] and sample objects merged. Mitochondrial content QC was carried out using a custom gene set (*ND1, ND2, ND3, ND4L, ND4, ND5, ND6, COX3,* and *CYTB*) and cells expressing more than 20% of mitochondrial genes were filtered out. Cells expressing fewer than 200 genes were also filtered out and doublets excluded using the scDblFinder package[99]. Cynomolgus macaque gene identifiers (*Macaca fascicularis*) were converted to rhesus macaque (*Macaca mulatta*) genes directly using the orthogene R package[108] (method = "gprofiler"); genes with non 1:1 mappings were filtered out (non121_strategy = "drop_both_species"). Data set normalization was performed using Seurat 4.0 with default settings and samples integrated using the Harmony package[109] with

default parameters, followed by clustering. UMAP plots and barplots were colored using viridislite[102] or batlow[103] scientific color map palettes.

**10× Visium HD CytAssist ST sample preparation, analysis, and cell type deconvolution**
Samples were processed as for 10× Visium CytAssist above, with the following modifications. Sequencing was completed on a NextSeq 2000 instrument at 275 million read pairs per fully-covered Capture Area. Library fastq files were aligned using spaceranger-3.0.1 software (10× Genomics).

Count matrices and associated tissue images were analyzed using Seurat (v5.1.0) in R (v4.4.0)[110]. A bin size of 8 μm was used as the resolution. Bins outside the tissue were identified using Loupe Browser (v8) and excluded from downstream analyses. The remaining bins were normalized using the NormalizeData function, variable features were identified using the FindVariableFeatures function (nfeatures = 3000), and data were scaled using the ScaleData function. The SketchData function was used to downsample the dataset to 50,000 cells (method = "LeverageScore"), followed by NormalizeData, FindVariableFeatures (nfeatures = 3000), ScaleData, RunPCA, FindNeighbors (dims = 1:50), FindClusters, and RunUMAP functions. Clusters were then projected back onto the full dataset using the ProjectData function (dims = 1:50).

Robust Cell Type Decomposition (RCTD) was used to deconvolve Visium HD bins, utilizing an scRNA-seq reference[111]. Macaque scRNA-seq genes were converted to human orthologs using Ensembl[112], with only one-to-one orthologs used for deconvolution. In addition, scRNA-seq clusters containing fewer than five cells were excluded. The create.RCTD function (CELL_MIN_INSTANCE = 10, counts_MIN = 3, UMI_MIN = 6, gene_cutoff = 0, gene_cutoff_reg = 0, CONFIDENCE_THRESHOLD = 3, fc_cutoff_reg = 0.5) was applied. The run.RCTD function (doublet_mode = "Doublet") was used for deconvolution. The deconvolution was first applied to the sketch assay and subsequently projected onto the full dataset.

**Statistics & reproducibility**
No statistical method was used to predetermine sample size. One technical replicate (ONPRC017_A) was excluded from the scRNA-Seq analyses as this technical replicate did not pass quality controls. The experiments were not randomized, and the investigators were not blinded to allocation during experiments and outcome assessment.

**Reporting summary**
Further information on research design is available in the Nature Portfolio Reporting Summary linked to this article.

## Data availability
The single cell RNA-sequencing datasets generated for the current study are available in the Gene Expression Omnibus (GEO) under accession number GSE263989. The spatial transcriptomics datasets are available on Zenodo [https://doi.org/10.5281/zenodo.15477421]. Published datasets for cynomolgus macaque samples were downloaded from GEO, including GSE160043[18] [https://www.ncbi.nlm.nih.gov/geo/query/acc.cgi?acc=GSE160043], GSE194264[61] [https://www.ncbi.nlm.nih.gov/geo/query/acc.cgi?acc=GSE194264] and GSE149629[20] [https://www.ncbi.nlm.nih.gov/geo/query/acc.cgi?acc=GSE149629] or Zenodo 6918355[16] [https://zenodo.org/records/6918355]. Source data are provided with this paper.

## Code availability
Code used for sample processing and analysis is provided in the following GitHub repository [https://github.com/ejscience/2025_Wamaitha_Rhesus_OvarianReserve]. No custom code was used to generate the results reported in this manuscript. Supplementary

Information is available for this paper. Correspondence and requests for materials should be addressed to Dr. Amander T. Clark (clarka@ucla.edu).

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

## Acknowledgements

The authors would like to thank Xinmin Li and staff of the UCLA Technology Center for Genomics & Bioinformatics, the UCLA Translational Pathology Core Laboratory, the UCLA BSCRC microscopy core, and the UCSF Genomics CoLab. This project was funded by NIH Grant R01 HD098278-01 (A.T.C., J.D.H.). Rhesus macaque tissue collection was supported by P51 OD011092. We would like to acknowledge Chelsea Naito at the ONPRC for technical assistance and the ONPRC Division of Comparative Medicine staff for supporting these studies, especially including the Surgical Services Unit, Time-Mated Breeding Services, and the Pathology Services Unit. These studies were also supported by the P51 OD011092-funded ONPRC Endocrine Technologies Core and the OHSU Histopathology Shared Resource. S.E.W. was supported by a University of California Presidents Postdoctoral Fellowship and a Young Investigator Award from the Iris Cantor-UCLA Women's Health Education & Research Center (NCATS UCLA CTSI Grant Number UL1TR001881). E.J.R. was supported by NIH Grant F31CA284719. D.J.L. was supported by NIH Grants R01GM122902 and R01ES023297.

## Author contributions

Conceptualization: A.T.C., J.D.H.; Investigation: S.E.W., E.J.R., E.S., A.M.H., K.O., M.C.; Formal analysis: F.M., E.J.R., F.-M.H., S.E.W.; Writing: S.E.W., A.T.C.; Supervision: A.T.C., J.D.H., J.S., D.J.L., M.M.; Funding acquisition: A.T.C., J.D.H., J.S., D.J.L.

## Competing interests

A.T.C. is on the board of the International Society for Stem Cell Research. The remaining authors declare no competing interests.
