## [Transparent Peer Review file · Nature Communications]

Defining the cell and molecular origins of the primate ovarian reserve

Corresponding Author: Professor Amander Clark

Version 0:

Reviewer comments:

Reviewer #2

(Remarks to the Author)

Overall, the authors have made important improvements to the manuscript. I feel the manuscript still finds challenges in reconciling missing data generation or developmental time points, which is understandable given to the precious nature of the samples. For example, the authors use visium for the challenging CS16 and CS20 samples but insights remain limited without matched scRNA-seq. These gaps in data give rise to speculation such as early pregranulosa cells give rise to fetal theca, which does not appear to be completely justified by the data analysis.

The authors have tried to address this, contextualising their data- particularly with inclusion of cross species datasets for biological insight and deeper insights into temporal pregranulosa patterns. However, there are still many questions left unanswered in this manuscript. It is important to note there is still no open access to data analysis scripts. Overall, this is an important dataset that is certainly of interest to the developmental biology community, but some key areas around interpretive clarity and novel biological insights are still left unaddressed.

It is essential to ensure that all necessary materials are made available for review. Preparing a GitHub repository to document the scripts and further detail the process in the Methods section will be important for replicability.

The authors have outlined the fetal tissue collection time points and their biological relevance, covering key stages of ovarian development from sex determination through primordial follicle formation and postnatal stages. These tissues do provide valuable insight into critical developmental transitions. Studying these time points within the tissue as opposed to follicles extracted from ovaries is also insightful

There is reasonable justification for the visium platform and localising specific regional gene expression. The authors address later limitations in performing scRNA-seq, matched datasets may have given further biological insight. In addition, the authors have corroborated their conclusions above with external datasets which strengthens their conclusions. Cross-species validation has strengthened their findings, and it is reassuring to see similar cell populations across datasets. It is plausible that it may not be possible to identify transitional cell types.

Thank you for the added information regarding spatial and temporal expression of FOXL2 and KRT19 in the rhesus ovary. The data gives interesting insight into pregranulosa cell dynamics – although to me it still seems unclear how this work and the identification of a PG 1.5 goes against a two-wave model, even if there is a later loss of spatial patterning.

The authors find pregranulosa cells drive early ovarian signatures which may be expected. I am a little unsure about any further conclusions the authors are trying to draw in this analysis lines 252 – 257.

The authors discuss this finding in lines 390 – 397/408 – 426. The authors describe a granulosa subtype (g7) expressing all theca (COL1A2, CYP17A1, DLK1, HSD3B2), granulosa (GATM, IFI6 and SERPINE2), and stromal NR2F2 markers. There is also expression of granulosa markers within the theca subclusters. I feel the reason for this is still unclear, at least for any concrete conclusions, and for the reasons given above by the author it may not be possible to suggest fetal theca originate from pregranulosa lineages.

Reviewer #3

(Remarks to the Author)

Dear authors,

You have modified your paper very extensively, to accommodate the extensive suggestions made by the other reviewers, willing to improve the quality of the manuscript. I am not sure the results regarding PG1/PG2 as well as theca cells originating from PG are very strong or convincing, but the datasets are worth publishing. The paper reads now much more clearly, the analysis has more depth and hence the relevance has increased considerably.

The colors used in Figure S6E and S6F are difficult to see.

After I provided my comments to the rebuttal (above), I was also asked to comment on the answers to comments by Reviewer #1:

- The issue of how the age of the samples was provided throughout the manuscript was changed by the authors as suggested by Reviewer #1.
- Several textual issues raised were also clarified by the authors.
- I agree with Reviewer #1 that the authors could/should be more careful suggesting that the rhesus monkey is a good model for human.
- In agreement with Reviewer #1 (see my own comments), I find some of the conclusions not well supported by the data and should be perhaps weakened. In any case, I find the datasets provided worth publishing.
- Finally, although the authors answer all the questions of Reviewer #1 in the rebuttal letter, there is no clear relation to many requested changes/clarification in the actual manuscript (see for example point 19, 20, 21, 22, 23, 24, 25 by Reviewer #1). I would suggest the authors included some to the justification provided in the revised manuscript text and please provide a clear reference to the lines in the modified manuscript text so we can read the modifications in context.
- When the authors do provide lines, I don't think the lines mentioned in the rebuttal match with the lines in the paper hence it is very difficult to match the two. However, having the two matched would very much facilitate further communication on specific points.

Version 1:

Reviewer comments:

Reviewer #2

(Remarks to the Author)

Response to review

Thank you to the authors for addressing the comments and making revisions. Also, the authors have now made all datasets and analysis code available. We note the changes made to clarify and soften interpretations that require further validation. These updates improve the manuscript's clarity, though some areas remain open for future investigation.

Reviewer #2

Overall, the authors have made important improvements to the manuscript. I feel the manuscript still finds challenges in reconciling missing data generation or developmental time points, which is understandable given to the precious nature of the samples.

Response: Thank you for the recognition of the challenges in working with NHP as a model and the precious nature of the samples.

No Comment

For example, the authors use visium for the challenging CS16 and CS20 samples but insights remain limited without matched scRNA-seq. These gaps in data give rise to speculation such as early pregranulosa cells give rise to fetal theca, which does not appear to be completely justified by the data analysis.

Response: We were unable to perform scRNA-Seq of ovaries at CS16 or CS20. We have amended the text to be more circumspect about the origin of fetal theca cells and propose that the origin of these cells still requires validation in future studies (lines 433 – 437).

This is a reasonable adjustment to the manuscript.

The authors have tried to address this, contextualising their data- particularly with inclusion of cross species datasets for biological insight and deeper insights into temporal pregranulosa patterns. However, there are still many questions left unanswered in this manuscript. It is important to note there is still no open access to data analysis scripts. Overall, this is an important dataset that is certainly of interest to the developmental biology community, but some key areas around interpretive clarity and novel biological insights are still left unaddressed.

Response: We have included a sentence in the discussion to highlight that there are many unanswered questions that we hope could be addressed in future studies (lines 551-553).

This is a reasonable adjustment to the manuscript.

It is essential to ensure that all necessary materials are made available for review. Preparing a GitHub repository to document the scripts and further detail the process in the Methods section will be important for replicability.

Response: A GitHub repository with the scripts has been created (https://github.com/ejscience/2025_Wamaita_Rhesus_OvarianReserve). The GEO has been released (GSE263989). Spatial transcriptomic data is available at Zenodo [10.5281/zenodo.15477421].

We have reviewed the GitHub repository which contains the R scripts used for analysis. (These are quite minimal). Data available through GEO Browser. Couldn't follow the Zenodo DOI but I am sure this will be made clear.

The authors have outlined the fetal tissue collection time points and their biological relevance, covering key stages of ovarian development from sex determination through primordial follicle formation and postnatal stages. These tissues do provide valuable insight into critical developmental transitions. Studying these time points within the tissue as opposed to follicles extracted from ovaries is also insightful

Response: Thank you for the comment.

No Comment

There is reasonable justification for the visium platform and localising specific regional gene expression. The authors address later limitations in performing scRNA-seq, matched datasets may have given further biological insight. In addition, the authors have corroborated their conclusions above with external datasets which strengthens their conclusions. Cross-species validation has strengthened their findings, and it is reassuring to see similar cell populations across datasets. It is plausible that it may not be possible to identify transitional cell types.

Response: This project started in 2019 with the sample collection, and Visium was the only whole transcriptome technology available at that time. If we were starting the project again in 2025, I agree, we would likely have planned things differently.

No Comment

Thank you for the added information regarding spatial and temporal expression of FOXL2 and KRT19 in the rhesus ovary. The data gives interesting insight into pregranulosa cell dynamics – although to me it still seems unclear how this work and the identification of a PG 1.5 goes against a two-wave model, even if there is a later loss of spatial patterning.

Response: In the discussion we will indicate that at W8, there are clearly two major types of granulosa cells and these are spatially patterned. The character of these cells, epithelial-like and mesenchymal-like are representative of the two waves of pre-granulosa cells in mice. However, a major difference is the absence of LGR5 in the epithelial like pregranulosa cells in nonhuman primates. Therefore, although it is possible that rhesus macaque ovary undergoes two waves of pre-granulosa cell specification it is less clear this involves two origins as demonstrated in the mouse (lines 512-515 and 524-525).

The authors are strongly encouraged to clearly distinguish between two conceptually distinct processes: the ontogeny of pre-granulosa cells (i.e., waves of specification) and the waves of primordial follicle activation (i.e., activated vs quiescent follicles). As they and others have noted, mice, humans, and non-human primates exhibit two spatially and temporally distinct waves of pre-granulosa cell specification. Whether these ontogenic waves give rise to functionally distinct follicle populations (e.g., early-activated vs long-term quiescent) is a separate matter. Conflating these two processes risks introducing conceptual confusion, especially for readers less familiar with ovarian developmental biology.

Regarding the ontogenic waves of pregranulosa cells, the presence or absence of a single marker (e.g., LGR5) is insufficient to define distinct pre-granulosa populations. Inferring one versus two ontogenic waves of pre-granulosa cells requires more granular, continuous spatio-temporal sampling of the fetal ovary than is currently presented. As the manuscript lacks samples between 8 and 15 weeks post-conception, its current dataset does not provide sufficient resolution to support or refute either the single- or dual-wave models. Moreover, the manuscript inaccurately implies inconsistency between non-human primate and human data regarding LGR5 expression. In humans, the second-wave pre-granulosa cells do not express LGR5, and LGR5 expression appears to be restricted to early supporting gonadal cells. Thus, the assertion that findings in non-human primates do not align with those in humans should be revised to reflect the current literature.

The authors find pregranulosa cells drive early ovarian signatures which may be expected. I am a little unsure about any further conclusions the authors are trying to draw in this analysis lines 252 – 257.

Response: We will adjust the conclusion to state that 'markers enriched in pregranulosa cells from the rhesus Visium data set including GATM, AMHR2, RGS5, MFGE8, and SERPINE2 were also enriched in the 10x cyno datasets, this was particularly notable at W6_2 (Fig S4G) (lines 256-258).

This is a reasonable adjustment to the manuscript

The authors discuss this finding in lines 390 – 397/408 – 426. The authors describe a granulosa subtype (g7) expressing all theca (COL1A2, CYP17A1, DLK1, HSD3B2), granulosa (GATM, IFI6 and SERPINE2), and stromal NR2F2 markers. There is also expression of granulosa markers within the theca subclusters. I feel the reason for this is still unclear, at least for any concrete conclusions, and for the reasons given above by the author it may not be possible to suggest fetal theca originate from pregranulosa lineages.

Response: We understand your concern, and therefore will adjust the language to indicate that although pregranulosa cell gene expression was identified in putative fetal theca cells, further studies are needed to understand the origin of these cells in the rhesus ovary (lines 433-437).

This is a reasonable adjustment to the manuscript

Response to review

Reviewer #2

Overall, the authors have made important improvements to the manuscript. I feel the manuscript still finds challenges in reconciling missing data generation or developmental time points, which is understandable given to the precious nature of the samples.

Response: Thank you for the recognition of the challenges in working with NHP as a model and the precious nature of the samples.

For example, the authors use visium for the challenging CS16 and CS20 samples but insights remain limited without matched scRNA-seq. These gaps in data give rise to speculation such as early pregranulosa cells give rise to fetal theca, which does not appear to be completely justified by the data analysis.

Response: We were unable to perform scRNA-Seq of ovaries at CS16 or CS20. We have amended the text to be more circumspect about the origin of fetal theca cells and propose that the origin of these cells still requires validation in future studies (lines 433 – 437).

The authors have tried to address this, contextualising their data- particularly with inclusion of cross species datasets for biological insight and deeper insights into temporal pregranulosa patterns. However, there are still many questions left unanswered in this manuscript. It is important to note there is still no open access to data analysis scripts. Overall, this is an important dataset that is certainly of interest to the developmental biology community, but some key areas around interpretive clarity and novel biological insights are still left unaddressed.

Response: We have included a sentence in the discussion to highlight that there are many unanswered questions that we hope could be addressed in future studies (lines 551-553).

It is essential to ensure that all necessary materials are made available for review.

Preparing a GitHub repository to document the scripts and further detail the process in the Methods section will be important for replicability.

Response: A GitHub repository with the scripts has been created (https://github.com/ejscience/2025_Wamaitha_Rhesus_OvarianReserve). The GEO has been released (GSE263989). Spatial transcriptomic data is available at Zenodo [10.5281/zenodo.15477421].

The authors have outlined the fetal tissue collection time points and their biological relevance, covering key stages of ovarian development from sex determination through primordial follicle formation and postnatal stages. These tissues do provide valuable insight into critical developmental transitions. Studying these time points within the tissue as opposed to follicles extracted from ovaries is also insightful

Response: Thank you for the comment.

There is reasonable justification for the visium platform and localising specific regional gene expression. The authors address later limitations in performing scRNA-seq, matched datasets may have given further biological insight. In addition, the authors have corroborated their conclusions above with external datasets which strengthens their conclusions. Cross-species validation has strengthened their findings, and it is reassuring to see similar cell populations across datasets. It is plausible that it may not be possible to identify transitional cell types.

Response: This project started in 2019 with the sample collection, and Visium was the only whole transcriptome technology available at that time. If we were starting the project again in 2025, I agree, we would likely have planned things differently.

Thank you for the added information regarding spatial and temporal expression of FOXL2 and KRT19 in the rhesus ovary. The data gives interesting insight into pregranulosa cell dynamics – although to me it still seems unclear how this work and the identification of a PG 1.5 goes against a two-wave model, even if there is a later loss of spatial patterning.

Response: In the discussion we will indicate that at W8, there are clearly two major types of granulosa cells and these are spatially patterned. The character of these cells, epithelial-like and mesenchymal-like are representative of the two waves of pre-granulosa cells in mice. However, a major difference is the absence of LGR5 in the epithelial like pregranulosa cells in nonhuman primates. Therefore, although it is possible that rhesus macaque ovary undergoes two waves of pre-granulosa cell specification it is less clear this involves two origins as demonstrated in the mouse (lines 512-515 and 524-525).

The authors find pregranulosa cells drive early ovarian signatures which may be expected. I am a little unsure about any further conclusions the authors are trying to draw in this analysis lines 252 – 257.

Response: We will adjust the conclusion to state that ‘markers enriched in pregranulosa cells from the rhesus Visium data set including *GATM*, *AMHR2*, *RGS5*, *MFGE8*, and *SERPINE2* were also enriched in the 10x cyno datasets, this was particularly notable at W6_2 (Fig S4G) (lines 256-258).

The authors discuss this finding in lines 390 – 397/408 – 426. The authors describe a granulosa subtype (g7) expressing all theca (*COL1A2*, *CYP17A1*, *DLK1*, *HSD3B2*), granulosa (*GATM*, *IFI6* and *SERPINE2*), and stromal *NR2F2* markers. There is also expression of granulosa markers within the theca subclusters. I feel the reason for this is still unclear, at least for any concrete conclusions, and for the reasons given above by the author it may not be possible to suggest fetal theca originate from pregranulosa lineages.

Response: We understand your concern, and therefore will adjust the language to indicate that although pregranulosa cell gene expression was identified in putative fetal theca cells, further studies are needed to understand the origin of these cells in the rhesus ovary (lines 433-437).

Reviewer #3 (Remarks to the Author):

Dear authors,

You have modified your paper very extensively, to accommodate the extensive suggestions made by the other reviewers, willing to improve the quality of the manuscript. I am not sure the results regarding PG1/PG2 as well as theca cells originating from PG are very strong or convincing, but the datasets are worth publishing. The paper reads now much more clearly, the analysis has more depth and hence the relevance has increased considerably.

Response: We appreciate your efforts in the first review to provide constructive feedback and agree that the manuscript was improved as a result.

The colors used in Figure S6E and S6F are difficult to see.

Response: We tried several color combinations, and this gave the best result. It is very difficult to see the orange (PGCs) because PGCs are extremely rare at this stage in development.

After I provided my comments to the rebuttal (above), I was also asked to comment on the answers to comments by Reviewer #1:

Response: You have my gratitude for serving as a double reviewer. I know that good reviews take considerable time, and I am sure like all of us you are extremely busy, so you have our thanks.

- The issue of how the age of the samples was provided throughout the manuscript was changed by the authors as suggested by Reviewer #1.

Response: We felt this was an excellent suggestion and made it easier to compare to other species (such as Cyno and human)

- Several textual issues raised were also clarified by the authors.

Response: Thank you

- I agree with Reviewer #1 that the authors could/should be more careful suggesting that the rhesus monkey is a good model for human.

Response: We will remove this sentence and instead of saying that we think the rhesus monkey is a good model for human, we will instead quote the study of van Wagenen and Simpson that has shown the sequence of gonadal developmental events in the rhesus macaque is comparable to the human (lines 65-67).

- In agreement with Reviewer #1 (see my own comments), I find some of the conclusions not well supported by the data and should be perhaps weakened. In any case, I find the datasets provided worth publishing.

Response: Thank you for his, the community also thinks the data is worth publishing. In response to requests for the release of the GEO after publishing the work in a preprint server, the data is now fully available to the scientific community.

- Finally, although the authors answer all the questions of Reviewer #1 in the rebuttal letter, there is no clear relation to many requested changes/clarification in the actual manuscript (see for example point 19, 20, 21, 22, 23, 24, 25 by Reviewer #1). I would suggest the authors included some to the justification provided in the revised manuscript text and please provide a clear reference to the lines in the modified manuscript text so we can read the modifications in context.

Response: In this response to review, we have highlighted (via Comment boxes in the Marked Changes version of the uploaded revised manuscript) where Reviewer 1's points from the original submission are addressed in the revised submission. These are also listed below (line numbers refer to the Marked Changes version – clean version just has changes accepted and comments resolved).

1 - 6, 9: Changed all references in manuscript from trimesters/days to weeks and amended Figure 1a.

7, 8, 14. Removed text from the manuscript.

10, 18. Included higher magnification images in Figure S2.

12. See lines 103 - 117.

13. See lines 163 - 166.

15. See lines 390 - 395.

16. See Fig. S7g

17. Amended to “granulosa within the medullary activated follicles” on line 445.

19. See lines 485 - 488; removed comparative text regarding cynomolgus macaque for clarity, but have now included the human cCasp3 data suggested by the reviewer, which we referenced in the response to the review but mistakenly omitted from the text.

20. See lines 466 - 467 regarding human hormone activity; “In human ovaries, CYP17A1 was only detected briefly between W19-24 and then after W33 – this latter timepoint correlates with our rhesus W19^{83,84}.” See also discussion lines 532 - 537 “Previous human and rhesus studies describe an increase in androgen / estrogen production by the ovary in later gestation; our rhesus data links this to 1) the W19 emergence of granulosa/theca cells in activated follicles, and 2) locates these hormonal units to the cortico-medullary region of the ovary. Data in the later stages of human ovarian development is rare, but the morphological, cellular and molecular similarities with the rhesus ovary suggest that these conclusions are likely applicable to humans as well.”

21. Removed text; amended our conclusions in lines 474 - 485 to say apoptotic germline cells are preferentially situated in early activated follicles at the cortico-medullary without implying causation linked to hormone activity.

22. See lines 323 - 328; lines 476 - 479.

23. See Figure S2B; also removed text.

24. See lines 438 - 473

25. See lines 435 - 437, and 474 - 484.

26. See response to point 21; see text in lines 45 - 49; 55 - 58; 347 - 353; 532 - 548.

27. See Figure 6; see line 347 - 353; removed text relating to follicle ovulation.

- When the authors do provide lines, I don't think the lines mentioned in the rebuttal match with the lines in the paper hence it is very difficult to match the two. However, having the two matched would very much facilitate further communication on specific points.

Response: We have now highlighted the line changes in the revised manuscript (see above) to avoid the problems with mismatches between rebuttal and manuscript.

Reviewer #2 (Remarks to the Author):

Dear reviewer #2: We are grateful to your suggestions as we work through the review process. Please find our responses to the second revision below (in green and called Revised response).

Response to review

Thank you to the authors for addressing the comments and making revisions. Also, the authors have now made all datasets and analysis code available. We note the changes made to clarify and soften interpretations that require further validation. These updates improve the manuscript's clarity, though some areas remain open for future investigation.

Revise response: Thank you for your comment, we also agree that areas remain open for future investigation.

Reviewer #2

Overall, the authors have made important improvements to the manuscript. I feel the manuscript still finds challenges in reconciling missing data generation or developmental time points, which is understandable given to the precious nature of the samples.

Response: Thank you for the recognition of the challenges in working with NHP as a model and the precious nature of the samples.

No Comment

Revise response: Acknowledged

For example, the authors use visium for the challenging CS16 and CS20 samples but insights remain limited without matched scRNA-seq. These gaps in data give rise to speculation such as early pregranulosa cells give rise to fetal theca, which does not appear to be completely justified by the data analysis.

Response: We were unable to perform scRNA-Seq of ovaries at CS16 or CS20. We have amended the text to be more circumspect about the origin of fetal theca cells and propose that the origin of these cells still requires validation in future studies (lines 433 – 437).

This is a reasonable adjustment to the manuscript.

Revise response: Thank you

The authors have tried to address this, contextualising their data- particularly with inclusion of cross species datasets for biological insight and deeper insights into temporal pregranulosa patterns. However, there are still many questions left unanswered in this manuscript. It is important to note there is still no open access to data analysis scripts. Overall, this is an important dataset that is certainly of interest to the developmental biology community, but some key areas around interpretive clarity and novel biological insights are still left unaddressed.

Response: We have included a sentence in the discussion to highlight that there are many unanswered questions that we hope could be addressed in future studies (lines 551-553).

This is a reasonable adjustment to the manuscript.

Revise response: Thank you.

It is essential to ensure that all necessary materials are made available for review. Preparing a GitHub repository to document the scripts and further detail the process in the Methods section will be important for replicability.

Response: A GitHub repository with the scripts has been created (https://github.com/ejscience/2025_Wamaitha_Rhesus_OvarianReserve). The GEO has been released (GSE263989). Spatial transcriptomic data is available at Zenodo [10.5281/zenodo.15477421].

We have reviewed the GitHub repository which contains the R scripts used for analysis. (These are quite minimal). Data available through GEO Browser. Couldn't follow the Zenodo DOI but I am sure this will be made clear.

A link has been created in the manuscript making the data uniquely findable in Zenodo and also linked to the GitHub (<https://zenodo.org/records/15477421>).

The authors have outlined the fetal tissue collection time points and their biological relevance, covering key stages of ovarian development from sex determination through primordial follicle formation and postnatal stages. These tissues do provide valuable

insight into critical developmental transitions. Studying these time points within the tissue as opposed to follicles extracted from ovaries is also insightful

No Comment

Revise response: Acknowledged

There is reasonable justification for the visium platform and localising specific regional gene expression. The authors address later limitations in performing scRNA-seq, matched datasets may have given further biological insight. In addition, the authors have corroborated their conclusions above with external datasets which strengthens their conclusions. Cross-species validation has strengthened their findings, and it is reassuring to see similar cell populations across datasets. It is plausible that it may not be possible to identify transitional cell types.

Response: This project started in 2019 with the sample collection, and Visium was the only whole transcriptome technology available at that time. If we were starting the project again in 2025, I agree, we would likely have planned things differently.

No Comment

Revise response: Acknowledged

Thank you for the added information regarding spatial and temporal expression of FOXL2 and KRT19 in the rhesus ovary. The data gives interesting insight into pregranulosa cell dynamics – although to me it still seems unclear how this work and the identification of a PG 1.5 goes against a two-wave model, even if there is a later loss of spatial patterning.

Response: In the discussion we will indicate that at W8, there are clearly two major types of granulosa cells and these are spatially patterned. The character of these cells, epithelial-like and mesenchymal-like are representative of the two waves of pre-granulosa cells in mice. However, a major difference is the absence of LGR5 in the epithelial like pregranulosa cells in nonhuman primates. Therefore, although it is possible that rhesus macaque ovary undergoes two waves of pre-granulosa cell specification it is less clear this involves two origins as demonstrated in the mouse (lines 512-515 and 524-525).

The authors are strongly encouraged to clearly distinguish between two conceptually distinct processes: the ontogeny of pre-granulosa cells (i.e., waves of specification) and the waves of primordial follicle activation (i.e., activated vs quiescent follicles). As they and others have noted, mice, humans, and non-human primates exhibit two spatially and temporally distinct waves of pre-granulosa cell specification. Whether these ontogenic waves give rise to functionally distinct follicle populations (e.g., early-activated vs long-term quiescent) is a separate matter. Conflating these two processes risks introducing conceptual confusion, especially for readers less familiar with ovarian developmental biology.

Revise response: we agree that the language around this in the discussion is challenging for the reader less familiar with ovarian development. Therefore, we have updated the last part of this paragraph by deleting the two sentences that conflate the two-wave hypothesis of PG specification from follicle type (deleted sentences from line 738).

Regarding the ontogenic waves of pregranulosa cells, the presence or absence of a single marker (e.g., LGR5) is insufficient to define distinct pre-granulosa populations. Inferring one versus two ontogenic waves of pre-granulosa cells requires more granular, continuous spatio-temporal sampling of the fetal ovary than is currently presented. As the manuscript lacks samples between 8 and 15 weeks post-conception, its current dataset does not provide sufficient resolution to support or refute either the single- or dual-wave models. Moreover, the manuscript inaccurately implies inconsistency between non-human primate and human data regarding LGR5 expression. In humans, the second-wave pre-granulosa cells do not express LGR5, and LGR5 expression appears to be restricted to early supporting gonadal cells. Thus, the assertion that findings in non-human primates do not align with those in humans should be revised to reflect the current literature.

Revise response: We have revised the results to embed the rhesus discoveries into the current human literature. Specifically, we have introduced the rhesus cognates of early somatic cells and early supporting gonadal cells (ESGCs) (lines 158-160, 195-196, 307-309, 736-737) and indicated when the first pre-granulosa cells (PG1) emerge in rhesus (W5-W6) (lines 315-316, 739-741 and model in Figure 6). We also have revised the results to introduce PG2 cells as being identified later, between W6-W8 (lines 476-477). We have also indicated that in agreement with human and cyno, PG1 cells are not a major contributor to quiescent follicle formation (line 561-563). Finally, we have also indicated that functional models will be needed to address whether establishment of quiescent follicles is related to position in the ovary rather than association with PG type (line 567-568).

We have corrected the LGR5 assertion (line 751-754) as follows: 'In addition, consistent with second wave preGC-II/PG2 cells in humans^{13,14}, LGR5 is also not expressed by second wave PG1.5 or PG2 cells in the rhesus ovary. This is distinct from the mouse where LGR5 is a major marker of second wave PG cells¹²'.

The authors find pregranulosa cells drive early ovarian signatures which may be expected. I am a little unsure about any further conclusions the authors are trying to draw in this analysis lines 252 – 257.

Response: We will adjust the conclusion to state that 'markers enriched in pregranulosa cells from the rhesus Visium data set including GATM, AMHR2, RGS5, MFGE8, and SERPINE2 were also enriched in the 10x cyno datasets, this was particularly notable at W6_2 (Fig S4G) (lines 256-258).

This is a reasonable adjustment to the manuscript

Revise response: Thank you.

The authors discuss this finding in lines 390 – 397/408 – 426. The authors describe a granulosa subtype (g7) expressing all theca (COL1A2, CYP17A1, DLK1, HSD3B2), granulosa (GATM, IFI6 and SERPINE2), and stromal NR2F2 markers. There is also expression of granulosa markers within the theca subclusters. I feel the reason for this is still unclear, at least for any concrete conclusions, and for the reasons given above by the author it may not be possible to suggest fetal theca originate from pregranulosa lineages.

Response: We understand your concern, and therefore will adjust the language to indicate that although pregranulosa cell gene expression was identified in putative fetal theca cells, further studies are needed to understand the origin of these cells in the rhesus ovary (lines 433-437).

This is a reasonable adjustment to the manuscript

Revise response: Thank you.